# Protein Sequence Design in a Latent Space via Model-based Reinforcement Learning

## Abstract

Proteins are complex molecules responsible for different functions in the human body. Enhancing the functionality of a protein and/or cellular fitness can significantly impact various industries. However, their optimization remains challenging, and sequences generated by data-driven methods often fail in wet lab experiments. This study investigates the limitations of existing model-based sequence design methods and presents a novel optimization framework that can efficiently traverse the latent representation space instead of the protein sequence space. Our framework generates proteins with higher functionality and cellular fitness by modeling the sequence design task as a Markov decision process and applying model-based reinforcement learning. We discuss the results in a comprehensive evaluation of two distinct proteins, GFP and His3, along with the predicted structure of optimized sequences using deep learning-based structure prediction.

## 1 Introduction

Proteins mediate the fundamental processes of cellular fitness and life. Iterated mutations on various proteins and natural selection during the biological evolution diversify traits, eventually accumulating beneficial phenotypes. Similarly, in protein engineering and design, the directed evolution of proteins has proved to be an effective strategy for improving or altering the proteins' functions or cellular fitness for industrial, research, and therapeutic applications (Yang et al., 2019; Huang et al., 2016). However, the protein sequence space of possible combinations of 20 amino acids is too large to search exhaustively in the laboratory, even with high-throughput screening from the diversified library (Huang et al., 2016). In other words, directed evolution becomes trapped at local fitness maxima where library diversification is insufficient to cross *fitness valleys* and access neighboring fitness peaks. Moreover, functional sequences in this vast space of sequences are rare and overwhelmed by nonfunctional sequences.

To tackle the limitations, data-driven methods have been applied to protein sequence design. They used reinforcement learning (RL) (Angermueller et al., 2019), Bayesian optimization (Wu et al., 2017; Belanger et al., 2019; Terayama et al., 2021; Stanton et al., 2022), and generative models (Jain et al., 2022; Kumar & Levine, 2020; Hoffman et al., 2022) in a model-based fashion, *i.e.*, using a protein functionality predictor trained on experimental data to model the local landscape. Despite the advances obtained by these methods, it is still challenging to generate optimized sequences that are experimentally validated. We suggest that the cause for this is two-fold. The first cause is that the optimization process is usually performed by generating candidate sequences directly through amino acid substitutions (Belanger et al., 2019) or additions (Angermueller et al., 2019). Given the vast search space, these methodologies are highly computationally inefficient and commonly lead to the exploration of parts of the space with a low chance of having functional proteins. In designing biological sequences, previous literature explored optimizing a learned latent representation space (Gómez-Bombarelli et al., 2018; Stanton et al., 2022). Similarly, in this paper we investigate the optimization of sequences via RL directly in a latent representation space rather than in the protein sequence space. Actions, *e.g.,* small perturbations in the latent vector, taken in this representation space can intuitively be understood as *walking* through a local functionality/fitness landscape.

The second cause is related to the models used as an oracle for optimization. These models are trained on experimental data obtained for a specific function, covering only a small portion of the

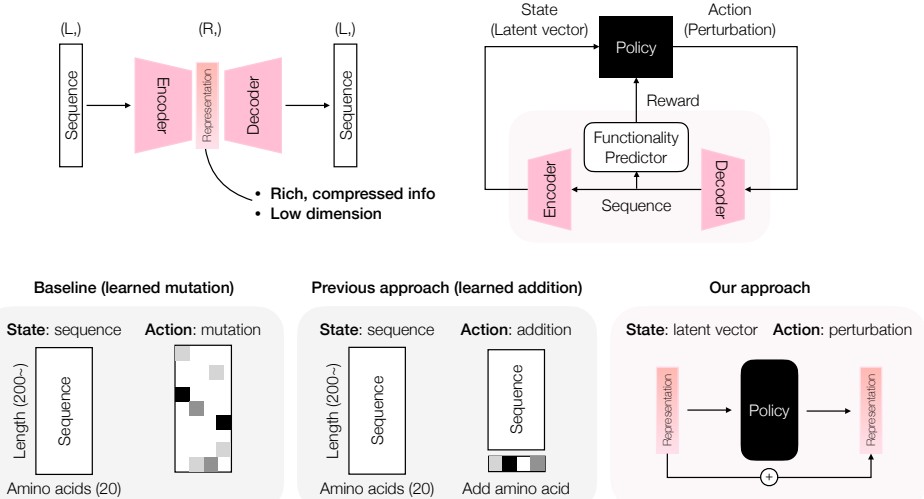

Figure 1: The framework's overview describes (top left) the encoder-decoder architecture trained to represent protein sequences in a latent space and (top right) the RL framework. The state is defined as the representation in the latent space, while the action is the perturbation in this representation. The perturbed representation is decoded back to a protein sequence using a sequence decoder. The reward is based on the functionality predicted by the functionality predictor (oracle). Bottom row shows three RL-based state and action modeling options.

protein space, and their accuracy is consequently restricted to this small region. Later, we will demonstrate that even the most advanced model-based optimizations can assign high functionality values for randomly generated protein sequences. These random sequences, however, are unlikely to be functional. To reduce false positives, we suggest augmenting the experimental data with random sequences (*i.e.*, negative examples) assigned to a low functionality value. Such augmentation also helps set boundaries around the experimental data distribution in which the oracle can be trusted.

We model protein sequence design as a Markov Decision Process (MDP) to optimize a latent representation. Our method trains the optimization policy using a model-based deep reinforcement learning (RL) framework (Fig. 1). At each timestep, the policy updates the latent representation by small perturbations to maximize protein functionality or cellular fitness, *i.e., walking* uphill through the local landscape until the end of an episode. We evaluate our method in two evaluation tasks, optimizing the functionality of the green fluorescence protein (GFP) and the cell fitness of imidazoleglycerol-phosphate dehydratase (His3). Our results show that the proposed framework design sequences with higher protein functionality and cellular fitness than existing methods. Ablation studies show that, based on the evaluation of various model options for state and action for the RL framework, the proposed latent representation update can successfully optimize the protein and search the vast design space. We provide visual evidence that the trained policy can traverse the local functionality landscape efficiently for GFP and His3. Our method is general and can also be applied to representations learned from protein structures, such as those presented in Zhang et al. (2022); Eguchi et al. (2022).

## 2  RELATED WORKS

**Protein representation learning**    Representation learning methods aim to learn compact and expressive features describing data. Since a protein is composed of a sequence of distinct amino acids (N=20), it can be interpreted as a large word in which each character is an amino acid. Due to the similarity with natural language processing (NLP) tasks, methods such as BERT (Devlin et al., 2018) and GPT-2 (Radford et al., 2019) have been used to train protein language models (Alley et al., 2019; Brandes et al., 2022; Lin et al., 2022; Ferruz et al., 2022; Rives et al., 2021). The protein language model trained with 250 million sequences by Rives et al. (2021) has shown to learn representations containing meaningful information about biological properties and reflecting protein structure. It was shown that the learned representation could be generalized across different applications achiev-

ing state-of-the-art results for supervised prediction of mutational effects. Representation learning has also been applied to learn protein structures (Zhang et al., 2022). An antibody-specific structure encoder was trained using variational autoencoders by Eguchi et al. (2022).

**Protein functionality prediction**   Similar to protein representation learning, for functionality prediction, methods inspired by NLP continue to achieve high performance. For the TAPE benchmark (Rao et al., 2019), long short-term memory (LSTM) (Hochreiter & Schmidhuber, 1997) and transformers (Vaswani et al., 2017) are the top performing methods. Notin et al. (2022) proposed a transformer-inspired architecture named Tranception for autoregressive functionality prediction that achieve state-of-the-art performance in 87 protein deep mutational scanning (DMS) datasets. Representing the protein as an image and using a convolutional neural network architecture (He et al., 2016) are also investigated in (Rao et al., 2019). Also proposed was the use of representations learned from a protein language model for zero-shot predictions (Alley et al., 2019; Meier et al., 2021). Utilizing protein structures as inputs, graph neural networks have been used to predict functionality in (Gligorijević et al., 2021). Finally, combining a sequence-based and structure-based representation for functionality prediction using language models and geometric vector perceptrons is proposed in (Wang et al., 2022).

**Biological sequence design**   Various machine learning methods have been used to design biological sequences. DynaPPO is a model-based RL algorithm proposed specifically for biological sequence design (Angermueller et al., 2019), which generates sequence from left to right by adding amino acids. Bayesian optimization was applied for cell fitness maximization in (Brookes et al., 2019; Belanger et al., 2019; Terayama et al., 2021; Swersky et al., 2020; Stanton et al., 2022). From these approaches, Stanton et al. (2022) used Bayesian optimization to optimize the sequence directly in a latent space by training a denoising autoencoder that learns representations for corrupted sequences. It is important to note that our approach, compared to Stanton et al. (2022), does not use the corrupt-and-denoise idea and defines optimization as an episodic task to use reinforcement learning. An evolutionary algorithm for efficient biological sequence optimization was proposed in (Sinai et al., 2020). Methods using generative models to search and sample optimized sequences are investigated in (Brookes et al., 2019; Kumar & Levine, 2020; Jain et al., 2022; Hoffman et al., 2022; Melnyk et al., 2021).

## 3   METHODOLOGY

This section describes the primary components of the model-based RL framework, depicted in Fig. 1. Details of network architectures are provided in Sec. 5 and Appendix A.

### 3.1   PROTEIN REPRESENTATION LEARNING

**Sequence Encoder**   Let $x \in \mathcal{X}$ be a (protein) sequence of length $L$, where $L$ denotes the number of amino acids in the sequence and $\mathcal{X}$ is the sequence space. Each element $x_i$ of the sequence is a discrete number representing one of 20 amino acids, and $i$ is the $i$-th element from $x$. Given $x$, the encoder network $e$ produces a representation $y$ of dimensions $(R, )$. The sequence encoder is divided into two parts: (i) a pre-trained protein language model encoder is used to obtain latent embeddings, (ii) a dimensionality reduction step is used to obtain the final representation.

In the first part, a pre-trained protein language model is leveraged (see Fig. 2(a)). We employ the ESM-2 model (Lin et al., 2022) trained on the Uniref50 dataset (Suzek et al., 2015). This model uses a BERT (Devlin et al., 2018) encoder and is trained with 150 million parameters in an unsupervised fashion (see Appendix A.1). ESM-2 is utilized to map mutation effects onto a latent space in our model. Given the sequence $x$ as an input, ESM-2 outputs a matrix $q \in \mathcal{Q}$ of dimensions $(L + 2, E)$, where $E$ is the dimension size of the embeddings, and $\mathcal{Q}$ is the embedding space. In ESM-2, a CLS (classification) token is prepended to the sequence, and an EOS (end-of-sequence) token is appended to the sequence when generating the embeddings. They serve as sequence-beginning and sequence-ending characters, respectively.

In the second part, the output of the encoder with dimensions $(L+2, E)$ is preprocessed before being passed to the policy. We use only the CLS token embedding to reduce the dimension from $(L+2, E)$ to $(E, )$. However, this could still be large for the action space given the embedding dimension $E$

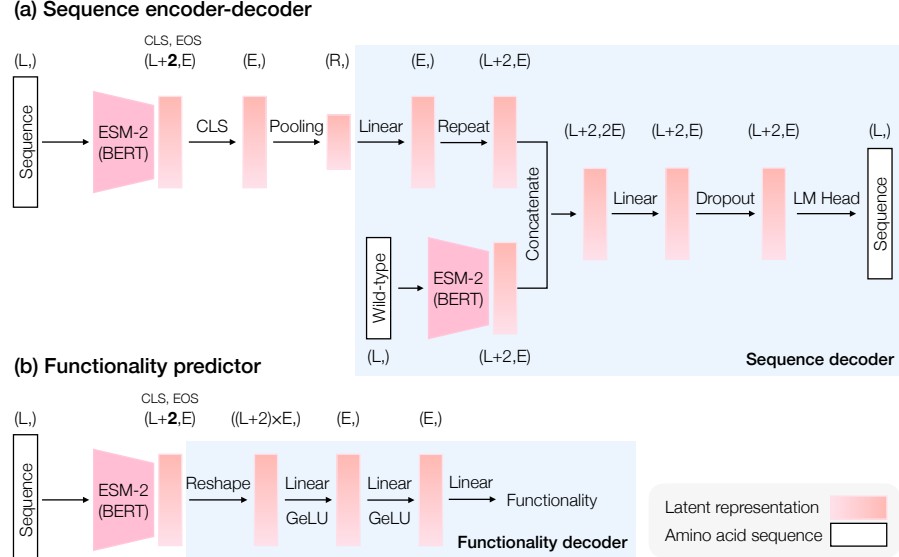

Figure 2: Model Architecture

of even the smallest ESM-2 model is $480$. We hence use 1-dimensional adaptive average pooling to further reduce the dimension to $(R, )$ and obtain the final representation.

**Sequence decoder** Given a reduced representation $y$ of size $(R, )$, we recover the amino acid sequence $x$ using the sequence decoder $d$. The representation size is expanded to $(E, )$ using a linear layer. To recover the original $(L + 2, E)$ dimensions of the embeddings obtained by ESM-2, the dimensions of the reduced representation are first expanded. Then, this matrix is concatenated with the wild-type embeddings obtained using ESM-2 of the pre-trained encoder of size $(L+2, 2E)$ followed by a linear layer. After passing to a dropout layer, the recovered representation is passed to the decoder's language model head, which maps the representation back to the sequence space, i.e., predicting each amino acid in the sequence from their associated embeddings. The output is the recovered sequence $x$.

### 3.2 PROTEIN FUNCTIONALITY PREDICTION

The functionality predictor network depicted in Fig. 2(b) is trained as a downstream task of the pre-trained ESM-2 encoder network. Given an input sequence $x$, the encoder network produces a representation $q \in \mathcal{Q}$ of dimensions $(L + 2, E)$. We flatten the representation to a vector of size $((L+2) \times E, )$ that is passed to an architecture with two linear layers followed by GeLU (Hendrycks & Gimpel, 2016) non-linear activation. The final linear layer predicts a value $k \in \mathbb{R}$, where $k$ is the functionality metric to be maximized and $\mathbb{R}$ is the set of real numbers representing possible values for this metric. While the encoder can be shared since we did not modify the architecture of the encoder nor the weights, the functionality decoder $f(x)$ is assumed to be trained independently for each task. To train the functionality predictor, negative examples are included when sampling the training batch to prevent the model from assigning high values to sequences outside the experimental data distribution. Negative examples are defined as random sequences with a low functionality/fitness value, and are a crucial data enhancement technique for reducing false positives. The augmentation can help set boundaries around the experimental data region that can be trusted.

### 3.3 PROTEIN SEQUENCE DESIGN VIA MODEL-BASED REINFORCEMENT LEARNING

The final component trains a policy using an off-policy RL algorithm that models the reward function based on the functionality predictor.

**System Modeling** Consider a policy $\pi$ that interacts with a fully observable environment during each episode. An episode is defined by the number of timesteps in which the policy takes actions

to optimize the target sequence in the latent space. We sample an initial state $s_0 \in \mathcal{Q}$ from the representation space (state space) at the beginning of each episode. At each timestep $t$, the agent observes a state $s_t \in \mathcal{Q}$ and selects an action $a_t \in \mathcal{A}$ according to a stochastic policy $\pi : \mathcal{Q} \to \mathcal{A}$, where $\mathcal{A}$ is the action space. The action $a_t$ is defined as a perturbation in which a continuous value $a_{t_j} \in [-\epsilon, \epsilon]$ is chosen for the $j$-th dimension, and $\epsilon$ is a hyperparameter that can be tuned based on the representation distribution of the experimental data. The hyperparameter $\epsilon$ controls how conservative the policy is when traversing the functionality or fitness landscape. Both $s_t$ and $a_t$ have $R$ dimensions.

We model the environment using a Markov Decision Process (MDP) and assume that the next state $s_{t+1}$ is conditioned only on the current state $s_t$ and action $a_t$. This function calculates the element-wise sum of $s_t$ and $a_t$ so that $s_{t+1} = s_t + a_t$. As a result of performing the action $a_t$, the agent receives the reward $r_t$. The reward is a function $r_t = r(s_t, a_t)$, where $r$ is the reward function $r : \mathcal{Q} \to \mathbb{R}$, and $\mathbb{R}$ is the set of real numbers representing reward values. This data is later used to train the policy using an off-policy reinforcement learning algorithm. The agent interacts with the environment until reaching a terminal state, which is one of the followings: (i) when the last timestep $T$ is reached; (ii) when $f(d(s_t))$ for the current state is greater than a hyperparameter threshold; and (iii) when $s_t$ is in a part of the representation space $\mathcal{Q}$ where $f(d(s_t))$ cannot be trusted. Condition (iii) states how far $s_t$ is from the experimental data. Each timestep's data is stored in a replay buffer as a 5-tuple $(s_t, a_t, s_{t+1}, r_t, m_t)$ where $m_t$ identifies the terminal state.

**Off-Policy Reinforcement Learning**  We further optimize the policy using Soft Actor Critic (SAC) (Haarnoja et al., 2018), an off-policy entropy-regularized reinforcement learning (RL) algorithm for continuous action spaces. Off-policy is a type of RL learning that learns to optimize independently of the agent's actions. SAC is trained to maximize future rewards and entropy to promote randomness in action space exploration. An optimal policy $\pi^*$ following this objective is given as

$$\pi^* = arg \max_\pi \mathbb{E}_\pi \left[ \sum_{t=0}^{\infty} \gamma^t (r(s_t, a_t) + \alpha H(\pi(\cdot|s_t))) \right], \tag{1}$$

where $\gamma \in [0, 1]$ is a discount factor, $\alpha$ is a temperature hyperparameter and $H$ is entropy. Details regarding SAC optimization and how actions are sampled by the policy $\pi$ are given in Appendix A.3.

## 4 RESULTS

### 4.1 EXPERIMENT SETUP

**Datasets**  We chose proteins of different lengths and functions for robustness in evaluation: the green fluorescent protein (GFP) that can be found naturally in jellyfish and imidazoleglycerol-phosphate dehydratase (His3) that is a key enzyme in our human body. The dataset proposed in (Sarkisyan et al., 2016) is used to train the GFP encoder-decoder and its functionality predictor. The dataset contains 54,025 mutant sequences, with log-fluorescence intensity associated with each sequence. The length $L$ of protein sequences is 237. For the His3 protein, we used the dataset in (Pokusaeva et al., 2019) that consists of mutant sequences of its evolutionarily-relevant segments and is associated with its growth rate. We processed the data to 50,000 sequences with the length of $L = 30$. This dataset is used to train the His3 encoder-decoder and fitness predictor. Datasets are randomly split into non-overlapping training and testing sets with a ratio of 90:10. The encoder, decoder, and functionality predictor use the aforementioned training set. The test set is used for evaluating the trained models and selecting initial states for optimization.

**Oracles**  We train two separate oracles for each dataset to prevent the circular use of the functionality predictor: one is exclusively used for optimization (which we call *optimization oracle*), and another is used to calculate the performance metric in experiments (which we call *evaluation oracle*). We follow the guideline presented in Kolli et al. (2022) that proposes the use of multiple oracles with different numbers of layers, activation functions, and hyperparameters to improve the reliability of the optimization process. We describe the experiment details and the comparison of evaluation and optimization oracles' performance in Appendices A.2 and A.7.

**Implementation Details**   The encoder and decoder use a pre-trained ESM-2 model (Lin et al., 2022) with 150 million trainable parameters. Optimization and evaluation oracles are trained using different ESM-2 models (Lin et al., 2022). The optimization oracle is trained using the model with 150 million parameters while the evaluation oracle uses the model with 35 million parameters. The latent representation space is set to $R = 8$, which sets the size of state and action vectors. The magnitude of the perturbation $\epsilon$ applied to each element of the action vector is set to $0.1$. The episode length $T$ is set to 20 for GFP and 10 for His3. We set three alternative experimental rewards: (1) a *dense* reward, defined as $r_t = f(d(s_t))$, (2) an *absolute* reward, defined as a binary value of $r_t = 1$, if $f(d(s_t)) > r_{th}$ and 0 if otherwise (values for $r_{th}$ are listed in Table 12), and (3) a *binary* reward, defined as a binary value of $r_t = 1$, if $f(d(s_t)) > f(d(s_{t-1}))$ and 0 if otherwise. To set the initial state $s_0$ of GFP, we sampled sequences with low (1.4 to 1.7) and high (3.2 to 3.4) functionality values from the test set. For His3, we sampled sequences with fitness values between 0.5 and 0.8. Appendix A describes other details regarding selecting initial states, network architectures, training hyperparameters, and optimization of our method and baseline methods.

**Evaluation Metrics**   We use three evaluation metrics as reported in Jain et al. (2022): performance, novelty, and diversity. We also consider two additional metrics for robustness: the originality of optimized sequences, named as *original*, i.e. sequences not contained in the training dataset, and the distance between the optimized sequence and wild-type, named as *dist(WT)*. The performance evaluation metric is calculated as the mean predicted functionality from the top $K$ generated sequences. The predicted functionality is obtained by using the evaluation oracle presented in Section 3.2. Let the generated sequences be contained in the following set $\mathcal{G}^* = \{g_1^*, \cdots, g_K^*\}$, performance is defined as $\frac{1}{K} \sum_i f(g_i^*)$. The novelty evaluation metric is defined to assess if the policy is generating sequences similar to the ones contained in the experimental data. Defining $\mathcal{P}$ as the experimental data set containing the wild-type protein sequence, novelty is given as follows:

$$\frac{1}{K \cdot |\mathcal{P}|} \sum_{g_i^* \in \mathcal{G}^*} \sum_{p_j \in \mathcal{P}} dist(g_i^*, p_j), \tag{2}$$

where $dist$ is defined as the number of different amino acids of two sequences. The diversity evaluation metric is defined as the mean of the number of amino acids that are different among the optimized sequences and is defined as:

$$\frac{1}{K(K-1)} \sum_{g_i^* \in \mathcal{G}^*} \sum_{g_j^* \in \mathcal{G}^* - \{g_i^*\}} dist(g_i^*, g_j^*). \tag{3}$$

The original metric is defined as $\frac{1}{K} \sum_i 1([g_i^* \notin \mathcal{P}])$ and the distance from wild-type (WT) metric is given as $\frac{1}{K} \sum_{g_i^* \in \mathcal{G}^*} dist(w, g_i^*)$, where $w$ is the wild-type sequence. When testing the protein functionality of GFP, we include the presence of the chromophore region (residues SYG in the wild-type protein) in the optimized sequence, as these residues are related to the ability to emit fluorescence.

## 4.2   EXPERIMENT

We generate 100 sequences for each method, and the 10 highest-performing sequences are evaluated.

**GFP Sequence Design**   The results obtained for GFP sequence design are described in Table 1. We compare with four optimization methods: CbAS (adaptive sampling) (Brookes et al., 2019), BO (bayesian optimization) (Swersky et al., 2020), GFlowNet (generative model) (Jain et al., 2022), and DynaPPO (reinforcement learning) (Angermueller et al., 2019). The term directed evolution refers to the average functionality values of initial states used in the optimization process. We also include random mutations of initial states. It is seen in Table 1 that the proposed method outperforms directed evolution. Only the proposed method and CbAS optimize GFP effectively, whereas BO, GFlowNet, and DynaPPO achieve low performance. Even though our method limits the action to a small step in the latent space, it gets higher novelty and diversity when compared to CbAS, which also limits the search space when optimizing. It was interesting to observe that two of the

GFP sequences optimized by the proposed method achieved higher predicted functionality values, 3.9705 and 3.8059, when compared to the experimental wild-type functionality value, 3.72. It is interesting to observe that methods that achieve the highest distance from the wild-type sequence are the ones that achieve the lowest performance.

| Model | Performance | Novelty | Original | dist(WT) | Diversity | Chromophore |
|---|---|---|---|---|---|---|
| Ours | **3.491** ± 0.352 | 8.451 | 100% | 7.700 | 6.311 | 100% |
| Directed evolution | 3.287 ± 0.237 | 7.704 | - | 6.849 | 4.858 | 100% |
| CbAS | 3.155 ± 0.153 | 7.712 | 80% | 6.900 | 1.956 | 100% |
| Random-1 | 2.824 ± 0.100 | 6.611 | 80% | 7.186 | 7.716 | 100% |
| Random-5 | 2.280 ± 0.275 | 13.91 | 100% | 9.950 | 12.37 | 90% |
| Random-P | 1.511 ± 0.797 | 14.71 | 100% | 14.15 | 14.62 | 100% |
| BO | 0.581 ± 0.095 | 36.96 | 100% | 36.70 | 6.867 | 100% |
| DynaPPO | 0.004 ± 0.003 | 218.9 | 100% | 219.3 | 224.1 | 0% |
| GFlowNet | 0.000 ± 0.002 | 199.4 | 100% | 200.1 | 12.53 | 0% |

Table 1: Results obtained for GFP sequence design. Random-1 and Random-5 indicates random mutations in 1 and 5 positions, respectively. Random-P indicates a random perturbation in the latent representation. Standard deviation for Novelty and dist(WT) are shown in Table 8.

**His3 Sequence Design**   Table 2 displays the results for His3, where our framework achieves the best performance. DynaPPO and BO were unable to optimize His3 effectively. Given that the length of the His3 protein is only 30, it is interesting to observe that a single random mutation is an effective strategy. Note, however, that even though the proposed method achieves higher overall performance, its novelty is lower than all other methods. This indicates that the decoder is recovering similar sequences within the representation space. The proposed method achieves higher novelty and diversity when compared to CbAS.

| Model | Performance | Novelty | Original | dist(WT) | Diversity |
|---|---|---|---|---|---|
| Ours | **0.945** ± 0.091 | 8.361 | 60% | 10.95 | 3.521 |
| CbAS | 0.749 ± 0.157 | 7.287 | 90% | 4.700 | 2.356 |
| Random-1 | 0.858 ± 0.058 | 7.372 | 80% | 7.350 | 7.716 |
| Random-5 | 0.678 ± 0.096 | 9.777 | 100% | 8.950 | 12.37 |
| Directed evolution | 0.616 ± 0.110 | 6.889 | - | 6.710 | 6.942 |
| DynaPPO | -0.201 ± 0.142 | 27.41 | 100% | 26.70 | 27.47 |
| BO | -0.313 ± 0.065 | 26.17 | 100% | 27.50 | 4.756 |

Table 2: Results obtained for His3 sequence design. Random-1 and Random-5 indicates random mutations in 1 and 5 positions, respectively. Standard deviation for Novelty and dist(WT) are shown in Table 9.

## 4.3 ABLATION STUDIES

**State and Action Modeling**   Table 3 shows how state and action modeling influence sequence design in GFP. For sequence modeling, we examine two types: a representation in which each amino acid is represented by one-hot encoding and a latent vector that is the encoder's output. We investigate three types of action modeling: (i) multi discrete sequence generation, (ii) conditional autoregressive addition of amino acids proposed in (Angermueller et al., 2019), and (iii) the proposed perturbation in the latent vector. Only the latent vector as the state and the perturbation as the action can optimize GFP and His3 tasks. Compared to the latent space learned based on the embeddings from ESM-2, using the one-hot encoded mutant sequence as input makes the identification of structural changes related to the protein's functional site challenging. Also, modeling the action as the addition of amino acids is ineffective for traversing the vast and sparse protein space.

**Representation Analysis**   We analyzed the influence of the optimization method on the latent vector as an input. Our method, which employs reinforcement learning for taking actions that update

| State | Action | GFP | His3 |
|---|---|---|---|
| Latent vector | Perturbation on latent vector | 3.491 ± 0.352 | 0.945 ± 0.091 |
| Directed evolution | | 3.287 ± 0.237 | 0.616 ± 0.110 |
| Sequence | Generate sequence | 0.006 ± 0.004 | -0.148 ± 0.043 |
| Latent vector | Generate sequence | 0.005 ± 0.003 | -0.139 ± 0.144 |
| Sequence | Amino acid addition | 0.004 ± 0.003 | -0.201 ± 0.142 |

Table 3: Comparison on the performance by the state and action modeling. For rows with the action defined as a mutation in the protein sequence, we train the policy using Proximal Policy Optimization (PPO) (Schulman et al., 2017) to handle the multiple discrete action space. More details regarding the state and action modeling for this ablation are presented in Appendix A.3.

a latent vector, is compared to random perturbation and to the aforementioned BO method proposed by Swersky et al. (2020), both of which take a latent vector as input. First, Swersky et al. (2020) performs better when using the latent vector input (2.601 ± 0.912) than when using the sequence as input (0.581 ± 0.095). It implies that the latent vector provides rich information in a low-dimensional space that is easier to optimize than the sequence space. Even with the same representation, our method outperforms Swersky et al. (2020) while achieving competitive novelty.

| Model | Performance | Novelty | Diversity | Chromophore |
|---|---|---|---|---|
| Ours | **3.491** ± 0.352 | 8.451 ± 2.05 | 6.311 | 100% |
| Directed evolution | 3.287 ± 0.237 | 7.704 ± 2.66 | 4.858 | 100% |
| Swersky et al. (2020) on latent space | 2.601 ± 0.912 | 8.077 ± 2.58 | 6.600 | 100% |
| Random perturbation | 1.511 ± 0.797 | 14.71 ± 5.90 | 14.616 | 100% |

Table 4: Comparison on our method and BO when using latent representation as an input.

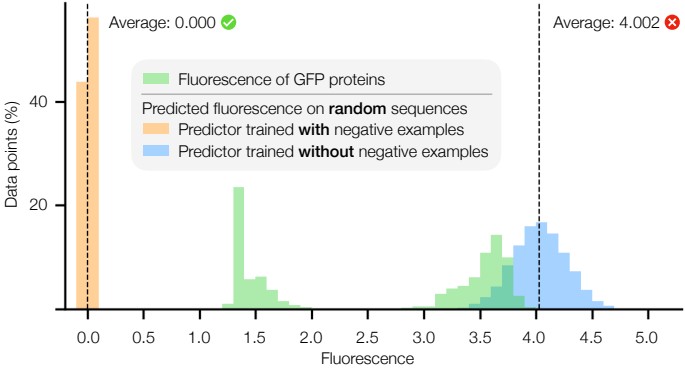

Figure 3: The predicted fluorescence of random sequences tested using a functionality predictor trained **with** or **without** negative examples.

**Training with Negative Examples** We trained the GFP functionality predictor with the original training dataset and the training dataset augmented with 40,000 negative examples (i.e., 82% of the size of the original training dataset). We tested both predictors with 10,000 negative examples that were not seen during training. Fig. 3 shows that the functionality predictor trained without negative examples incorrectly predicts a high value for non-functional sequences (mean=4.002), whereas the one trained with negative examples accurately assigns zero functionality (mean=0.0) to random non-functional sequences.

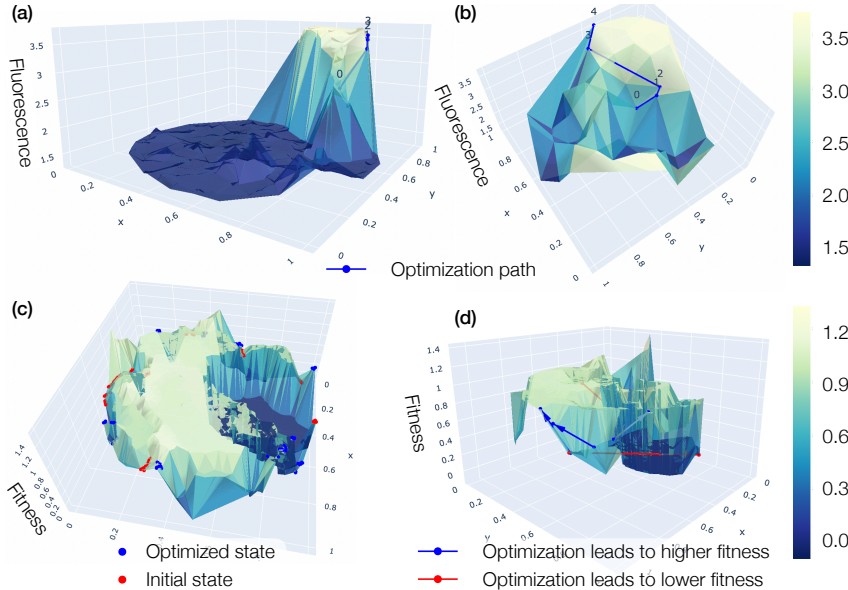

Figure 4: Visualization of the climbing process through a local functionality and fitness landscape. **(a) and (b)** Resulting optimization carried out by the trained policy during a single episode for the GFP task. The z-axis represents the intensity of log fluorescence. In (b) it is seen the policy traversing local optima due to the RL-based formulation of the proposed method. **(c)** Fitness landscape for the His3 dataset. We annotated initial states (red) and optimized states (blue). **(d)** Optimization performed by the trained policy for the His3 task. The representation space is reduced to two using t-SNE (Van der Maaten & Hinton, 2008). Appendix A.9 describes the methodology used to generate these landscapes. An HTML-formatted interactive version of these graphs can be found in the supplementary materials.

## 5 DISCUSSION

**How the trained policy traverses the functionality landscape** We qualitatively evaluate our trained policy by analyzing its ability to traverse the local landscape. Figs. 4 (a) and (b) visualize the optimization trajectory during one episode of GFP, indicating continued improvement of the fluorescence level. This demonstrates that the model can successfully *climb up* the local landscape throughout the episode. The policy can also traverse local optima thanks to the MDP formulation (See Appendix A.11). For His3, the cell fitness landscape of the entire dataset is shown in Fig. 4(c), which interestingly depicts a valley region that corresponds to part of the space leading to low fitness. Also, we can observe that the optimized states are scattered throughout the landscape, implying that our policy can design diverse proteins. Fig. 4(d) presents optimization steps taken by the trained policy, which now shows that large (optimistic) perturbations often lead to failure in optimization.

## 6 CONCLUSION

This paper addressed the problem of optimizing protein functionality and cellular fitness using data-driven methods. We investigated the limitations of the latest model-based biological sequence design methods and proposed a novel optimization framework that can efficiently traverse the latent representation space, as opposed to optimizing the protein sequence by mutations in its amino acid sequence. Our framework trained a policy that continually updates the latent representation throughout an episode. We modeled the problem as an MDP to maximize future rewards, which allowed the agent to explore the landscape while learning an optimal policy. This special mechanism was critical to efficiently traversing the local functionality and fitness landscape. The proposed framework outperformed other baseline methods in performance and is competitive in terms of novelty. We have also demonstrated that a trained policy can navigate a local landscape effectively and traverse local optima. In the future, we intend to filter the generated sequences and conduct wet lab experiments to validate the proposed framework.

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

# A APPENDIX

## A.1 SEQUENCE ENCODER-DECODER

Next, we give more information regarding the architecture and the training process of the sequence encoder-decoder. The model architecture is seen in 2(a). The ESM-2 encoder comprises a token embedding layer and 30 transformer layers. Each transformer layer consists of multi-head attention followed by a layer normalization layer and two fully connected layers with GeLU (Hendrycks & Gimpel, 2016) non-linear activation. Language model head comprises of linear layer followed with GeLU activation and the linear layer using the weight of the embedding layer in the encoder. The number of attention heads is set to 20. We used the pre-trained weights provided in (Lin et al., 2022). The ESM-2 encoder weights are not updated during the fine-tuning process. The sequence encoder-decoder is fine-tuned using a masked language model objective. We define the loss function as the mean of cross entropy loss on mutated positions and the mean cross-entropy loss on non-mutated positions to ensure that the decoder also focuses on recovering mutated positions. Since we provide the wild-type representation during sequence recovery and mutants in the GFP dataset have a maximum of 15 mutations, such a loss function is important to prevent the decoder from predicting every sequence as the wild-type. We train the model to the train set of each dataset for 16 epochs using the Adam optimizer (Kingma & Ba, 2014). The initial learning rate of Adam is set to 8e-5, with weight decay set to 1e-5. The learning rate is reduced by $0.8\times$ every epoch. Table 5 shows the performance of a sequence decoder by the chosen embedding size of $R$.

| Dataset | Embedding size | Top-1 Accuracy | |
| --- | --- | --- | --- |
| | | Mutated positions | Non-mutated positions |
| GFP | 8 | 0.463 | 0.9958 |
| | 32 | 0.4762 | 0.9903 |
| His3 | 8 | 0.6439 | 0.8407 |
| | 32 | 0.8194 | 0.9022 |

Table 5: Performance of the sequence decoder for GFP and His3.

## A.2 FUNCTIONALITY PREDICTOR

Functionality and fitness predictors for GFP and His3 are also based on the pre-trained ESM-2 protein language model. The ESM-2 encoder has same architecture as described in Appendix A.1, and the decoder is described in Sec. 3. We fine-tuned the predictor on the train set of each dataset for 16 epochs using the Adam (Kingma & Ba, 2014) optimizer with the initial learning rate set to 8e-5 and weight decay set to 1e-5. The ESM-2 encoder weights are not updated during the fine-tuning process. The negative samples for GFP and His3 are associated with a functionality/fitness value of 0. The GFP value of 0 denotes an extremely low fluorescence level ($10^{(0-3.72)} = 0.0002$ times the brightness of the wild type protein). The His3 value of 0 denotes the minimum fitness value in the dataset, ranging from $[0, 1.63]$. We used the mean squared error between empirically obtained log-fluorescence intensity and predicted fluorescence as the loss function for GFP and the empirically obtained fitness and predicted fitness as the loss function for His3.

### A.2.1 OPTIMIZATION ORACLE

In Table 6 we report the test Spearman's $\rho$ and mean squared error (MSE) of the optimization oracle used for model-based optimization.

### A.2.2 EVALUATION ORACLE

In Table 7 we report the test Spearman's $\rho$ and mean squared error (MSE) of the evaluation oracle used for model-based optimization. Compared to the optimization oracle, the evaluation oracle utilize a different pre-trained ESM-2 model which also contains a different number of parameters. The evaluation oracle and optimization oracle are trained with similar methodology.

| Task | Test set | | Negative test set |
| --- | --- | --- | --- |
| | Spearman's $\rho$ | MSE | MSE |
| GFP | 0.8426 | 0.1436 | 2.930e-6 |
| His3 | 0.6635 | 0.0080 | 0.0164 |

Table 6: Performance of the optimization oracle for GFP and His3

| Task | Test set | | Negative test set |
| --- | --- | --- | --- |
| | Spearman's $\rho$ | MSE | MSE |
| GFP | 0.8320 | 0.4359 | 6.247e-05 |
| His3 | 0.6820 | 0.0110 | 0.0288 |

Table 7: Performance of the evaluation oracle for GFP and His3

### A.2.3    Training CbAS and DynaPPO with and without negative samples

We conducted an ablation study by training two baseline optimization methods, CbAS and DynaPPO, using both functionality predictors. The optimized sequences are then visualized using predicted structures obtained by AlphaFold2. As shown in Fig. 5, CbAS managed to propose a valid fluorescent protein when guided by an oracle trained with negative examples but failed when guided by an oracle trained without negative examples. Even though the structure predicted for the CbAS model for this case seems unstable, the oracle trained without negative examples assigned a very high log-fluorescence intensity value (4.014) for the sequence. Therefore, we argue that biological sequence design should be guided and evaluated by an oracle trained with an experimental dataset augmented with negative examples.

### A.3    Soft Actor Critic (SAC)

Recapitulating, an optimal policy $\pi^*$ following the SAC objective is given as

$$\pi^* = arg \max_\pi \mathbb{E}_\pi \left[ \sum_{t=0}^{\infty} \gamma^t (r(s_t, a_t) + \alpha H(\pi(\cdot|s_t))) \right], \tag{4}$$

where $\gamma \in [0, 1]$ is a discount factor, $\alpha$ is a temperature hyperparameter and $H$ is entropy. Following a policy $\pi$ the Q function or action-value function is defined as

$$Q^\pi(s, a) = \mathbb{E}_\pi \left[ \sum_{t=0}^{\infty} \gamma^t r(s_t, a_t) + \alpha \sum_{t=1}^{\infty} \gamma^t H(\pi(\cdot|s_t)) \right], \tag{5}$$

where the entropy is added from the first timestep. The Bellman equation for $Q^\pi$ at a timestep $t$ is given by

$$Q^\pi(s_t, a_t) = \mathbb{E}_\pi \left[ r(s_t, a_t) + \gamma(Q^\pi(s_{t+1}, \tilde{a}_{t+1}) + \alpha H(\pi(\cdot|s_{t+1}))) \right], \tag{6}$$

in which he state $s_{t+1}$ is used from the replay buffer and $\tilde{a}_{t+1}$ is sampled from the current policy. During training, SAC models three networks: a policy $\pi_\theta$ and two Q-functions $Q_{\phi 1}$ and $Q_{\phi 2}$. The

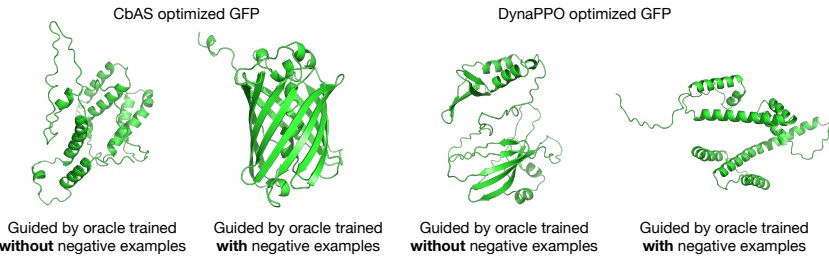

Figure 5: The structure of optimized sequences without and with negative examples.

two Q functions are due to the clipped double-Q trick, more details can be found in Haarnoja et al. (2018). The loss functions to train $Q_{\phi 1}$ and $Q_{\phi 2}$ in SAC are defined as

$$L(\phi_i, \mathcal{D}) = \mathbb{E}_{(s_t, a_t, r_t, s_{t+1}, m_t) \sim \mathcal{D}} \left[ (Q_{\phi_i}(s_t, a_t) - y(r_t, s_{t+1}, m_t))^2 \right], \tag{7}$$

where the target $y(r_t, s_{t+1}, d_t)$, substituting the entropy function, is given by

$$y(r_t, s_{t+1}, m_t) = r_t + \gamma(1 - m_t) \left( \min_{j=1,2} Q_{\phi_{targ,j}}(s_{t+1}, \tilde{a}_{t+1}) - \alpha \log \pi_\theta(\tilde{a}_{t+1}|s_{t+1}) \right), \tag{8}$$

where $\tilde{a}_{t+1} \sim \pi_\theta(\cdot|s_{t+1})$. The policy $\pi_\theta$ is trained to maximize the value function of state $s_t$. The value function $V^\pi$ is as follows

$$V^\pi(s_t) = \mathbb{E}_{a_t \sim \pi} \left[ Q^\pi(s_t, a_t) - \alpha \log \pi(a_t|s_t) \right]. \tag{9}$$

The policy is reparameterized and then optimized following Haarnoja et al. (2018). Using a squashing Gaussian function, sampling the action is finally defined as

$$\tilde{a}_{t_\theta}(s_t, \xi) = tanh(\mu_\theta(s_t) + \sigma_\theta(s_t) \odot \xi), \qquad \xi \sim \mathcal{N}(0, I). \tag{10}$$

## A.4 Implementation details

### A.4.1 Policy

**Soft Actor-Critic (SAC)** We used three fully connected layers with a hidden size set to 256 for both the actor and critic networks of SAC. The Adam optimizer Kingma & Ba (2014) is used for both actor and critic networks. The entropy temperature $\alpha$ is automatically tuned during training following Haarnoja et al. (2018).

**Proximal Policy Optimization (PPO)** For the ablation studies in Table 3 using the mutation in the protein sequence as output, we use Proximal Policy Optimization (PPO) Schulman et al. (2017) to handle the discrete action space setting. The PPO architecture is similar to the one used for the actor of SAC, using three fully connected layers with a hidden size set to 256. We define the action as a multiple of discrete actions for the third and fifth row in Table 3.

### A.4.2 Random Perturbation Baseline

We added a random value between $[-1, 1]$ to the initial state representation to test the effect of random perturbations on performance. Then, we decoded the sequence using the decoder presented in Sec. 3.1 and predicted the functionality using the functionality predictor presented in Sec. 3.2. Note that the baseline and random performance reported in the Tables 1 and 2 are also calculated on the top K candidates.

### A.4.3 Previous methods

For the comparison with previous baseline methods, we use the FLEXS implementation (Sinai et al., 2020) of DynaPPO, CbAS, and BO. All the models are trained for ten rounds. During training, we observed the performance saturation after the first few rounds. The sequences proposed in the last round are used to evaluate the baseline models. For DynaPPO, its ensemble model is trained for ten rounds, and the environment batch size is set to 10. For CbAS, a CNN architecture with a hidden size equal to 100 and a number of channels equal to 32 is used. Also, the generator block of CbAS is trained for ten epochs.

## A.5 Initial States of the optimization Process

We selected initial states for the optimization process in our framework according to two criteria: (i) sequences with room to improve, (ii) sequences sampled from regions that we can trust the oracle. Since the distribution for the ground truth functionality of GFP is bimodal, we sampled two sets of initial states from each mode, low initial states, and high initial states. For His3, only one set of initial states is chosen. As shown in Fig. 6, we sample the sequences from pre-determined regions that do not include the most functional mutants. Additionally, the sequences are sampled from examples in which the difference between their ground truth and predicted functionality or fitness is less than a threshold set to 0.1.

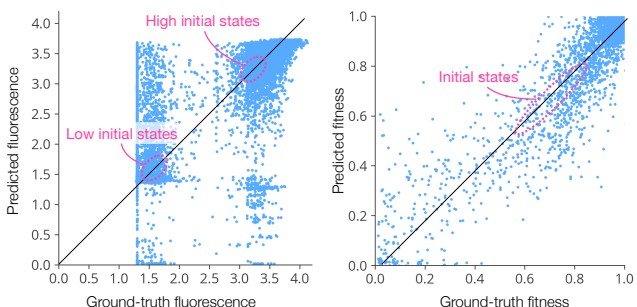

Figure 6: Ground truth and predicted fluorescence of GFP and fitness of His3.

## A.6 Evaluation on GFP and His3 design

In this section, along with the results presented in Tables 1 and 2 we report results on GFP and His3 on Novelty and dist(WT) with standard deviation.

| Model | Novelty | dist(WT) |
|---|---|---|
| Ours | 8.451 ± 2.05 | 7.700 ± 0.78 |
| Directed evolution | 7.704 ± 2.66 | 6.849 ± 1.90 |
| CbAS | 7.712 ± 2.05 | 6.900 ± 0.83 |
| Random-1 | 6.611 ± 1.02 | 7.186 ± 2.03 |
| Random-5 | 13.91 ± 1.22 | 9.950 ± 0.87 |
| Random-P | 14.71 ± 5.90 | 14.15 ± 5.76 |
| BO | 36.96 ± 5.51 | 36.70 ± 5.31 |
| DynaPPO | 218.9 ± 2.63 | 219.3 ± 2.37 |
| GFlowNet | 199.4 ± 2.00 | 200.1 ± 1.87 |

Table 8: Results obtained for GFP sequence design. Random-1 and Random-5 indicates random mutations in 1 and 5 positions, respectively. Random-P indicates a random perturbation in the latent representation.

| Model | Novelty | dist(WT) |
|---|---|---|
| Ours | 8.361 ± 2.01 | 10.95 ± 1.32 |
| CbAS | 7.287 ± 1.57 | 4.700 ± 0.64 |
| Random-1 | 7.372 ± 1.56 | 7.350 ± 1.39 |
| Random-5 | 9.777 ± 1.63 | 8.950 ± 1.36 |
| Directed evolution | 6.889 ± 1.57 | 6.710 ± 1.57 |
| DynaPPO | 27.41 ± 1.12 | 26.70 ± 1.19 |
| BO | 26.17 ± 1.03 | 27.50 ± 0.50 |

Table 9: Results obtained for His3 sequence design. Random-1 and Random-5 indicates random mutations in 1 and 5 positions, respectively.

## A.7 Comparison of evaluation and optimization oracle

In Tables 10 and 11, we compare results on GFP and His3 design using evaluation oracle and optimization oracle. It is shown that, for all methods, a decrease in performance using the evaluation oracle is observed. This decline was particularly pronounced for BO (Swersky et al., 2020). It is also observed that the standard deviation of the results increases when using the evaluation oracle.

| Model | Evaluation oracle | Optimization oracle |
|---|---|---|
| Ours | 3.491 ± 0.352 | 3.531 ± 0.06 |
| Directed evolution | 3.287 ± 0.237 | 3.370 ± 0.013 |
| CbAS (Brookes et al., 2019) | 3.155 ± 0.153 | 3.328 ± 0.044 |
| Random mutation (N=1) | 2.824 ± 0.100 | 3.410 ± 0.094 |
| Random mutations (N=5) | 2.280 ± 0.275 | 2.354 ± 0.522 |
| Random perturbation | 1.511 ± 0.797 | 1.973 ± 0.832 |
| BO (Swersky et al., 2020) | 0.581 ± 0.095 | 1.231 ± 0.034 |
| DynaPPO (Angermueller et al., 2019) | 0.004 ± 0.003 | 0.014 ± 0.001 |
| GFlowNet (Jain et al., 2022) | 0.000 ± 0.002 | 0.017 ± 0.000 |

Table 10: Comparison of the results obtained for GFP sequence design using evaluation and optimization oracle.

| Model | Evaluation oracle | Optimization oracle |
|---|---|---|
| Ours | 0.945 ± 0.091 | 0.961 ± 0.050 |
| CbAS (Brookes et al., 2019) | 0.749 ± 0.157 | 0.889 ± 0.092 |
| Random mutation (N=1) | 0.858 ± 0.058 | 0.856 ± 0.070 |
| Directed evolution | 0.616 ± 0.110 | 0.756 ± 0.013 |
| Random mutations (N=5) | 0.678 ± 0.096 | 0.518 ± 0.184 |
| DynaPPO (Angermueller et al., 2019) | -0.201 ± 0.142 | -0.067 ± 0.053 |
| BO (Swersky et al., 2020) | -0.313 ± 0.065 | -0.089 ± 0.029 |

Table 11: Comparison of the results obtained for His3 sequence design using evaluation and optimization oracle.

## A.8 ABLATION STUDY ON INITIAL STATE AND REWARD MODELING

Table 12 reports the effect of the initial state and reward. Note that, for this ablation study, the performance metric is computed using the optimization oracle. Starting from a state of low or high functionality, our framework optimizes performance relative to the directed evolution. The best performance is obtained when starting from a state of high functionality and training with an absolute reward. Our framework can generate novel sequences even when beginning with a high functionality value. Nonetheless, beginning from a state with low functionality, the trained policy does not produce designs with high functionality. This could mean that there may be a gap between regions with low functionality and regions with high functionality in the representation space that requires additional time steps to be explored in each episode.

| Initial state | Reward | Performance* | Novelty | Diversity |
|---|---|---|---|---|
| Low | Dense | 1.856 ± 0.631 | 70% | 8.032 |
| | Absolute ($r_{th} = 1.8$) | 1.855 ± 0.601 | 50% | 5.653 |
| | Binary | 1.546 ± 0.271 | 100% | 6.347 |
| | Directed evolution | 1.492 ± 0.008 | - | 5.426 |
| High | Dense | 3.448 ± 0.094 | 100% | 5.700 |
| | Absolute ($r_{th} = 3.3$) | 3.531 ± 0.06 | 100% | 6.311 |
| | Binary | 3.452 ± 0.081 | 95% | 5.279 |
| | Directed evolution | 3.370 ± 0.013 | - | 4.858 |

Table 12: Ablation studies investigating the sampling of initial states and the reward modeling for the GFP dataset. The policy trained with absolute reward and starting from high initial states is used for comparisons in Tables 1 and 3. Performance(*) is computed using the optimization oracle.

| Figure | Task | Number of | | Dimension reduction |
| --- | --- | --- | --- | --- |
| | | Representations | Sequences | |
| Fig. 4 (a) | GFP | 1.55M | 1403 | t-SNE |
| Fig. 4 (b) | GFP | 1.17M | 46 | t-SNE |
| Fig. 4 (c)(d) | His3 | 6.43K | 6434 | t-SNE |
| Fig. 8 | GFP | 1.17M | 46 | PCA |

Table 13: Details of the complete functionality landscape.

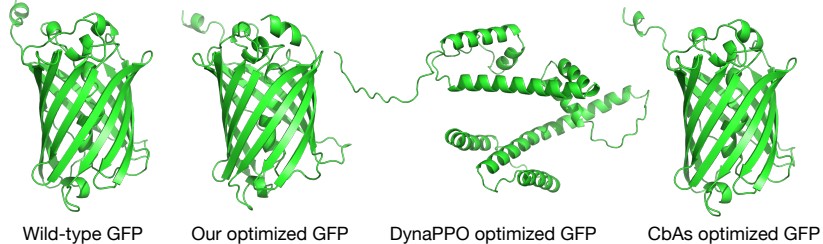

Wild-type GFP    Our optimized GFP    DynaPPO optimized GFP    CbAs optimized GFP

Figure 7: Structures predicted by AlphaFold2 for sequences optimized by the proposed method, DynaPPO, and CbAs for GFP.

## A.9    METHODOLOGY TO OBTAIN THE COMPLETE FUNCTIONALITY LANDSCAPE

The methodology to plot the Fig. 4 and Fig. 8 is detailed next. First, a range based on the representations obtained during the episode being plotted is defined. Then we decode the distinct sequences from the large number of representations using the decoder presented in Sec. 3.1. Table 13 shows the number of representations and sequences used to plot the complete functionality and fitness landscape. After this step, the reward is decoded for each sequence using the optimization oracle presented in Sec. 3.2. To reduce the dimensionality, t-SNE (Van der Maaten & Hinton, 2008) is applied in Fig. 4 and Principal Component Analysis (PCA) (F.R.S., 1901) is applied in Fig 8 with the two principal components kept to create the local landscape.

## A.10    ALPHAFOLD2 PREDICTIONS OF OPTIMIZED SEQUENCES

Fig. 7 visualizes the structure of the optimized sequence of GFP based on AlphaFold2 (Jumper et al., 2021). The sequences produced by our method and CbAS maintain the critical chromophore region that is known for emitting fluorescence in GFP and the beta sheets that secure the chromophore region. In contrast, DynaPPO-optimized sequences failed to preserve these essential structures of GFP. The different structures obtained by two reinforcement learning methods highlight the significance of state and action modeling in the design of biological sequences. Learning from a single-hot sequence encoding makes it difficult for an algorithm to identify crucial structural information. On the other hand, the latent vector trained by extracting information from a language model trained with millions of protein sequences can efficiently learn mutational effects and reflect protein structures, as shown in (Rives et al., 2021).

### A.11 POLICY ABILITY TO TRAVERSE LOCAL OPTIMA

We qualitatively evaluate our trained policy for the GFP by analyzing its ability to traverse local optima of the functionality landscape. The optimization process during one episode is shown in Fig. 8. Until the end of the episode, at timestep $t = 10$, the policy can traverse the landscape and maximize functionality efficiently. At timestep $t = 3$, it can escape local optima that do not trigger the conditions for the end of an episode. During timesteps $t = 7, 8, 9$, it can be seen that the policy is still maximizing the functionality metric until the end of the episode. Additional details and the methodology used to obtain the functionality landscape in Fig. 8 are explained in A.9.

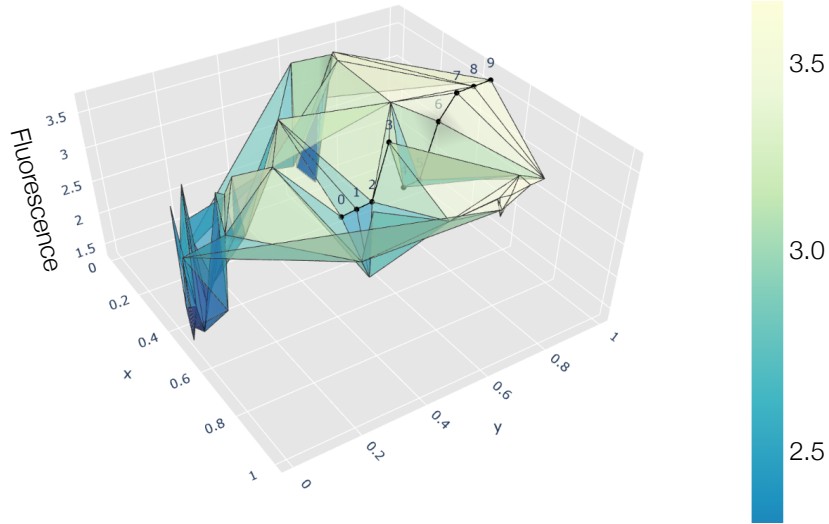

Figure 8: Optimization process performed by the trained policy during one episode for the GFP task. The x-axis and y-axis are the two principal components of the representation space calculated using Principal Component Analysis (F.R.S., 1901). The z-axis is the log-fluorescence intensity.

