# OpenReview forum: "Protein Sequence Design in a Latent Space via Model-based Reinforcement Learning"
_ICLR.cc/2023/Conference — Submitted to ICLR 2023_

### Official Review · Reviewer_62rv · 2022-10-22

**Confidence:** 5
**Correctness:** 2
**Technical Novelty And Significance:** 3
**Empirical Novelty And Significance:** 3
**Recommendation:** 3

**Clarity, Quality, Novelty And Reproducibility:**

The paper contains some novel ideas combining latent space optimization and RL. Unfortunately, the description of the method is rather unclear and the Appendix does not provide much clarity. After multiple readings of the method, I'm not 100% sure I understand the exact sequence of steps in their method architecture due to ambiguous language and contradictory statements. The reproducibility across a diverse set of tasks is a minor weakness in this paper compared to other MBO methods in the literature. This includes the modification of the benchmark in the literature which seems to drastically change the qualitative results compared to those previously seen.

Clarity:
- Include in the related work: Gomez-Bombarelli et al (2018) as using property prediction on the latent space for design was first explored by them.
- "2-dimensional vector representation" usually implies a 2x1 vector rather than a L x E matrix which is perhaps a bit confusing.
- "We recover the amino acid sequence from the reduced representation of size (R,). Using a linear
layer, the size of the representation is expanded to (E,). To recover the L dimension, we concatenate the reduced representation with the wild-type representation of the pre-trained encoder of size (L, E)," Is the reduced representation in this case (R,) here as stated in the first sentence? Otherwise, its not clear how you concat shape (L,E) with (R,), so I assume its the lifted representation (E,)? Then the following linear layer has weights (L+1, L)? This bit is confusing and seemingly the crux of the method.
- "Given an input sequence x, the predictor returns a value k ∈ R, where k is the functionality metric to be maximized and R is the set of real numbers representing possible values for this metric. k is predicted from a representation q with dimensions (L, E), and the functionality predictor f(q)." Figure 1 seems to imply that the input to the property predictor is in X (one-hot encoded) space since it feeds the input of the encoder into the functionality predictor but the text implies that the output of the encoder with representation (L,E) is the input to the functionality predictor.
- " The next state st+1 is sampled from a transition probability function p(st+1|st, at).... This function calculates the elementwise sum of st and at so that st+1 = st + at." So that its not really sampled? This seems like a confusing
- "Both st and at have E dimensions....The latent representation space is set to R = 8, which sets the size of state and action vectors" Are these vectors in E space or R space then?
- For the negative samples, how was the evaluation for Fig 2 done? How was the train/test split chosen since a random one likely doesn't model OOD effects?

Typos:
- "CbAsoptimize"-> "CbAS optimize" on Page 5

**Strength And Weaknesses:**

Strengths:
- There are a nice set of ideas presented in this work. Combining latent space optimization and reinforcement learning is an interesting set of ideas.
- Negative sampling-based data augmentation is a great idea that is likely a huge driver in improvements of this method as well as can be applied to most other MBO methods
- Sanity checking with AF2 is a nice

Weaknesses:
- As stated in Evaluation Metrics section, its problematic to evaluate performance with the same oracle that you use to do property prediction. I suggest following Kolli et al (2022)[1] to use an oracle trained on the full dataset for evaluation and using a smaller dataset for train/test split for functionality predictors.
- Are the random mutations truly random? If so, a stronger baseline would be to compete against random mutations ranked by an oracle.
- How were hyperparameters chosen on-line? Such as the action space, f(s_t) threshold, part of representation space that can't be trusted, the number of negative samples, the data distribution of negative samples, etc. Often methods in the Model-based optimization space can't be trusted if they have many hyperparameters without a methodology as to how to tune them off-line.
- It seems that the performance of some of these baseline methods largely departs from other instances such as in Jain et al (2022) perhaps due to implementation details or differences in value K. Can this be commented on (either via an ablation over K or validating or commenting on the irreproducibility of experiments in prior work)? This would have a huge impact on ranking of these MBO algorithms seems rather poor despite using the same benchmark.
- Why was the subset of two proteins chosen? We see in other works large variability in method ranking across methods and so they attempt to combat this by testing across 5+ benchmark tasks (occasionally in Notin et al (2022) [2] it has risen up to 87 DMS datasets for prediction tasks).
- The analysis in Fig 3 would be more convincing if a held-out oracle was used rather than a similar evaluation vs prediction oracle.


[1] Kolli et al (2022). "Data-Driven Optimization for Protein Design: Workflows, Algorithms and Metrics"
[2] Notin et al (2022). "Tranception: protein fitness prediction with autoregressive transformers and inference-time retrieval"

**Summary Of The Paper:**

This paper combines ideas from latent space optimization and reinforcement learning to propose a method for model-based optimization for biological sequences. They also introduce a nice idea for using random samples as negative sampling (data augmentation) which seems to significantly drive performance. They then apply their results to GFP and His3 and demonstrate improved performance when tested against a prediction oracle.

**Summary Of The Review:**

This paper presents some nice ideas(RL, latent space optimization, negative sampling) for an exciting problem settings. They have some interesting evaluations(qualitative analysis of climbing the landscape, sanity check against AF2). However, the primary weaknesses of this paper are twofold: (1) the exposition surrounding the method is ambiguous(ie seemingly inconsistent language) and the experimental section largely deprioritizes the explaining the core details that give someone working in this space confidence that the method is effective (eg hyper-parameter optimization, how train/test splits are performed, inconsistent results from the literature) and (2) the experiments don't follow best practices in the field that allow the reader to interpret the results at face value. Luckily, I think both of these issues can be rectified and are not detrimental to the idea itself.

---

> ### Author Response · Authors · 2022-11-12
> **Response to reviewer 62rv [1/5]**
>
> The authors are incredibly thankful to the reviewers for their insightful and detailed comments that helped improve our work. Please find below our response to each comment. We would be happy to follow up on any additional feedback.
>
> The authors appreciate the reviewer's critique of the unclear language in the Methodology and Appendix A.1 and A.2 sections. In the revised manuscript, we tried to conduct a comprehensive review. We rewrote the corresponding sections to address the reviewer's concerns and enhance the clarity of the experimental design. Please see the updated version of the Methodology, Appendix A.1, Appendix A.2, and Fig. 5 at [https://bit.ly/3hBeS9E](https://bit.ly/3hBeS9E)
>
> > **Comment #1 (Oracle)** As stated in Evaluation Metrics section, its problematic to evaluate performance with the same oracle that you use to do property prediction. I suggest following Kolli et al (2022)[1] to use an oracle trained on the full dataset for evaluation and using a smaller dataset for train/test split for functionality predictors.
> >
>
> Thank you for pointing out this critical issue. This point is very important since there might exist a leakage of information if we use the same oracle for optimization and evaluation. This concern was shared by other reviewers as well.
>
> We truly appreciate the suggestion to apply the method in (Kolli et al, 2022) and utilize multiple oracles. This study suggests training multiple oracles with different numbers of layers, activation functions, and hyperparameters. We followed the advice and trained another oracle from scratch for the GFP dataset using an ESM model that uses different network parameters (35M). This oracle was not used for optimization.
>
> As a result, we could utilize two oracles: (i) an optimization oracle (trained using the ESM model with 150M parameters) which is used in the current work and (ii) an evaluation oracle (trained using the ESM model with 35M parameters). Please find the evaluation results based on this new oracle below:
>
> |  | Evaluation oracle | Optimization oracle |
> | --- | --- | --- |
> | Spearman’s Rho | 0.7426 | 0.8426 |
> | MSE | 0.2872 | 0.1436 |
> | Ours | 3.491 ± 0.352 | 3.531 ± 0.06 |
> | Baseline | 3.287 ± 0.237 | 3.370 ± 0.013 |
> | CbAS (Brookes et al., 2019) | 3.155 ± 0.153 | 3.328 ± 0.044 |
> | BO (Swersky et al, 2020) | 0.581 ± 0.095 | 1.231 ± 0.034 |
>
> **Table** - Results obtained for GFP sequence design
>
> |  | Evaluation oracle | Optimization oracle |
> | --- | --- | --- |
> | Spearman’s Rho | 0.6820 | 0.6635 |
> | MSE | 0.0110 | 0.0080 |
> | Ours | 0.945 ± 0.091 | 0.961 ± 0.050 |
> | Baseline | 0.616 ± 0.110 | 0.756 ± 0.013 |
> | CbAS (Brookes et al., 2019) | 0.749 ± 0.157 | 0.889 ± 0.092 |
> | BO (Swersky et al, 2020) | -0.313±0.065 | -0.089 ± 0.029 |
>
> **Table** - Results obtained for His3 sequence design
>
> Summarizing the results, we find that all methods show a decrease in performance using the new oracle. This decline was particularly pronounced for BO(Swesky et al, 2020). We also note that the standard deviation of the results increases with the new evaluation oracle.
>
> Our method continues to outperform all baselines under the new oracle. In addition to the results above, we examined the functionality values of top-K sequences generated by the proposed method. Two of the identified sequences had high functionality values of 3.9705265 and 3.8059266, indicating that they were 1.78 and 1.20 times brighter than the value for the wild type protein, respectively. In the future, we plan to evaluate these sequences through wet lab experiments.
>
> Thank you once again for this important feedback, and we will add the new results to the main paper for GFP and His3 datasets.
>
> > **Comment #2 (Random Mutations)** Are the random mutations truly random? If so, a stronger baseline would be to compete against random mutations ranked by an oracle.
> >
>
> Regarding random mutations, for a fair evaluation, we generate random sequences in the same manner as optimization methods. First, we generate a number of random sequences, and then we compare the top K sequences as determined by the oracle. We hope this answers your question, and please let us know if there are any follow-up questions.
>
> > **Comment #3 (Hyperparameters)** How were hyperparameters chosen on-line? Such as the action space, f(s_t) threshold, part of representation space that can't be trusted, the number of negative samples, the data distribution of negative samples, etc. Often methods in the Model-based optimization space can't be trusted if they have many hyperparameters without a methodology as to how to tune them off-line.
> >
>
> Thank you for the clarification question. These are essential details that should be included in the paper. Please see below for a description of the key hyperparameters.

---

> ### Author Response · Authors · 2022-11-12
> **Response to reviewer 62rv [2/5]**
>
> | Hyperparameter | Tuning Methodology |
> | --- | --- |
> | Action space (embedding size) | For the action space, we performed ablation studies using the following values **[4, 8, 16, 32]**. The **choice of this parameter is a tradeoff between two factors**. The first factor is the **accuracy of the sequence decoder** (to recover the optimized sequence). The highest the action space, the more accurate the sequence decoder tends to be. The second factor is the **dimensionality of the action space**, as continuous control reinforcement learning algorithms usually are optimized in a low-dimensional space (less than 12 dimensions) [1]. The lower the action space, the better for training the optimization policy. |
> | Maximum length of the episode | We recommend choosing a value between **[5, 10, 15, 20], depending on the length of the target protein and the narrowness of the functionality/fitness landscape**. In this case, we define Narrowness as the average functionality/fitness by the number of positions mutated. For example, the landscape of GFP is narrow [2], indicating that the majority of GFP mutants with 10 mutated positions are non-fluorescent. The perturbation in the action space can result in zero, one, or multiple mutations in the decoded sequence. For GFP, the best results were obtained using a maximum length set to 20 (amino acid length of 237). For His 3, the maximum length was set to 10 (amino acid length of 30). It is also important to mention that we have an additional terminal condition for an episode, as stated in the manuscript: “when st is in a part of the representation space Q where f(s_t) cannot be trusted.” |
> | Range of the action (-r, r) | For the definition of the range of the action, we first calculate the maximum difference in each component of the learned representation in the training dataset. We then define **2 * r * maximum length of the episode = maximum difference in each component of representation.** Our objective using this definition was to ensure that the agent can sufficiently traverse the representation space. |
> | Reward function threshold | For choosing the reward function threshold we use the following methodology. We performed ablation testing the following values **[median functionality of initial states, 75% functionality of initial states, maximum functionality of initial states].** For GFP (in which we sample from both high functional and low function states) if the initial states are sampled from high functional states, the threshold is set as the median (Table 4, high). If the initial states are sampled from low functional states, the threshold is set as the maximum of initial states (Table 4, low). We reason that this choice leads to better results because generating an optimized sequence over the threshold is harder when the initial state is already a high functional sequence. This value needs to be tuned given the characteristics of the functional/fitness landscape, that directly affects exploration during the training of the RL algorithm. |
> | Number of negative examples | We set the number of negative examples in GFP and His3 experiments to 40,000. A guideline to determine this number is to select a value close to the number of positive examples in the experimental data set. |
> | Space Q where the oracle can be trusted | This space controls the terminating condition of an episode. An episode is terminated when the distance between the representation of the generated sequence and the representation of the wild type protein exceeds **a * (maximum distance from WT in the experimental dataset)**. In our work, we set a to 2 for GFP dataset and we set a to 1 for His3 dataset. When searching for the best value for a, the designer should consider **the accuracy of the sequence decoder and the length of the protein sequence** in the dataset to be optimized. |
>
> [1] Lillicrap, Timothy P., et al. "Continuous control with deep reinforcement learning." arXiv preprint arXiv:1509.02971 (2015).
>
> [2] Sarkisyan, Karen S., et al. "Local fitness landscape of the green fluorescent protein." Nature 533.7603 (2016): 397-401.

---

> ### Author Response · Authors · 2022-11-12
> **Response to reviewer 62rv [3/5]**
>
> > **Comment #4** **(Performance of Baseline Methods)** It seems that the performance of some of these baseline methods largely departs from other instances such as in Jain et al (2022) perhaps due to implementation details or differences in value K. Can this be commented on (either via an ablation over K or validating or commenting on the irreproducibility of experiments in prior work)? This would have a huge impact on ranking of these MBO algorithms seems rather poor despite using the same benchmark.
> >
>
> Thank you for sharing this important observation. An ideal situation would be to have a general oracle that is used to benchmark MBO algorithms so that when comparing performance, we can refer to the results in any papers.
>
> In our research, we wanted to investigate the use of negative samples and retrain the oracle for specific datasets. This means implementing and retraining all baselines for experiments. We utilized the official implementation codes and details of the cited work. We plan to make all codes available for reproducibility.
>
> Our experiment demonstrated that an oracle could significantly alter the ranking of these methods. We also observed that it was difficult to reproduce specific results, particularly those obtained through generative methods. In the future, we hope to develop a set of trustworthy oracles that can be used for optimization and evaluation, thereby enhancing the reproducibility and comparability of MBO methods.
>
> > **Comment #5 (Datasets)** Why was the subset of two proteins chosen? We see in other works large variability in method ranking across methods and so they attempt to combat this by testing across 5+ benchmark tasks (occasionally in Notin et al (2022) [2] it has risen up to 87 DMS datasets for prediction tasks).
> >
>
> Thank you for asking this question. We realize that we had omitted the description of this selection process in the paper. The two benchmarks were selected using the following criteria.
>
> **Condition 1**: The maximum number of mutation should be at least 3 (there is more than single mutation or pair mutation) [Only 5 out of 87 satisfies in DMS datasets]
>
> **Condition 2**: One should be able to build a reliable oracle (i.e., with a Spearman’s rho of 0.6 or above)  [Only 18 out of 87 satisfies in DMS Datasets]
>
> This leaves with only two protein datasets in the DMS datasets. More details regarding the datasets filtered by these criteria is shown in the tables below.
>
> | 1 mutation | 2 mutations | >3 mutations |
> | --- | --- | --- |
> | 76 | 6 | 5 |
>
> **Table** - Maximum number of mutations of 87 DMS dataset in Notin et al.
>
> | Dataset | Spearman’s Rho (Ensemble Tranception & EVE) | No. mutants |
> | --- | --- | --- |
> | **GFP_AEQVI_Sarkisyan_2016** | **0.706** | **51714** |
> | **HIS7_YEAST_Pokusaeva_2019** | **0.600** | **496137** * |
> | F7YBW8_MESOW_Aakre_2015 | 0.439 | 9192 |
> | CAPSD_AAV2S_Sinai_substitutions_2021 | 0.419 | 42328 |
> | GCN4_YEAST_Staller_induction_2018 | 0.272 | 2638 |
>
> **Table** - Spearman’s Rho and number of mutants of the datasets that have >3 mutations. We use GFP_AEQVI_Sarkisyan_2016 and HIS7_YEAST_Pokusaeva_2019 (top 2) for the experiments performed in our manuscript. *We used fourth segment of His3 of length 30 which is about 50000 mutants.
>
> Similarly, Jain et al (2022) reported their performance on three tasks (GFP, TF-Bind-8, AMP), with the length of TF-Bind-8 dataset being 8 amino acids. Angermueller et al (2019) reported their performance in three tasks in which the maximum length of amino acids is 50. One of the tasks optimized in (Angermueller et al, 2022) is a length-8 DNA sequence.
>
> Nonetheless, we concur with the reviewer that expanding the benchmark dataset and enhancing the robustness of the result would be beneficial. In the future, we would plan to expand our evaluation to include F7YBW8_MESOW_Aakre_2015, CAPSD_AAV2S_Sinai_substitutions_2021 dataset, and an additional antibody dataset.

---

> ### Author Response · Authors · 2022-11-12
> **Response to reviewer 62rv [4/5]**
>
> > **Comment #6 (Fig. 3)** The analysis in Fig 3 would be more convincing if a held-out oracle was used rather than a similar evaluation vs prediction oracle.
> >
>
> Thank you for suggesting this relevant research. We find it to be interesting and relevant to our research. We revised our manuscript as follows:
>
> In page 1, we include the previous work by Gomez-Bombarelli et al (2018) in the Introduction section as: “To tackle this problem in biological sequence design, previous literature (Gomez et al, 2018; Stanton et al, 2022) explored performing the optimization directly in a learned latent representation space. In this paper, we investigate the optimization of sequences via Reinforcement Learning in a latent representation space rather than in the protein sequence space”.
>
> > **Comment #7 (Related Work)** Include in the related work: Gomez-Bombarelli et al (2018) as using property prediction on the latent space for design was first explored by them.
> >
>
> Thank you for suggesting this relevant research. We find it to be interesting and relevant to our research. We revised our manuscript as follows:
>
> In page 1, we include the previous work by Gomez-Bombarelli et al (2018) in the Introduction section as: “To tackle this problem in biological sequence design, previous literature (Gomez et al, 2018; Stanton et al, 2022) explored performing the optimization directly in a learned latent representation space. In this paper, we investigate the optimization of sequences via Reinforcement Learning in a latent representation space rather than in the protein sequence space”.
>
> > **Comment #8** "2-dimensional vector representation" usually implies a 2x1 vector rather than a L x E matrix which is perhaps a bit confusing.
> >
>
> Thank you for the correction. We have modified our manuscript accordingly.
>
> > **Comment #9** "We recover the amino acid sequence from the reduced representation of size (R,). Using a linear layer, the size of the representation is expanded to (E,). To recover the L dimension, we concatenate the reduced representation with the wild-type representation of the pre-trained encoder of size (L, E)," Is the reduced representation in this case (R,) here as stated in the first sentence? Otherwise, its not clear how you concat shape (L,E) with (R,), so I assume its the lifted representation (E,)? Then the following linear layer has weights (L+1, L)? This bit is confusing and seemingly the crux of the method.
> >
>
> Thanks for the critical comment. We tried our best to rewrite the description regarding the sequence decoding from the representation with size (R,). The anonymized updated version can be seen in this link: [https://bit.ly/3hBeS9E](https://bit.ly/3hBeS9E)
>
> > **Comment #10** "Given an input sequence x, the predictor returns a value k ∈ R, where k is the functionality metric to be maximized and R is the set of real numbers representing possible values for this metric. k is predicted from a representation q with dimensions (L, E), and the functionality predictor f(q)." Figure 1 seems to imply that the input to the property predictor is in X (one-hot encoded) space since it feeds the input of the encoder into the functionality predictor but the text implies that the output of the encoder with representation (L,E) is the input to the functionality predictor.
> >
>
> Thank you again for pointing out this mistake. The input of the functionality predictor block is the sequence that is then passed through the ESM-2 encoder to get the embeddings. This is clarified through new figures and the updated description in the Methodology and Appendix. The anonymized updated version can be seen in this link: [https://bit.ly/3hBeS9E](https://bit.ly/3hBeS9E)
>
> > **Comment #11** "The next state st+1 is sampled from a transition probability function p(st+1|st, at).... This function calculates the elementwise sum of st and at so that st+1 = st + at." So that it's not really sampled? This seems confusing.
> >
>
> We recognize that the current statement could be confusing. The subsequent state is calculated by the element-wise sum, so it is not sampled. The sentence is re-written in the revised manuscript.

---

> ### Author Response · Authors · 2022-11-12
> **Response to reviewer 62rv [5/5]**
>
> > **Comment #12** "Both st and at have E dimensions....The latent representation space is set to R = 8, which sets the size of state and action vectors" Are these vectors in E space or R space then?
> >
>
> These vectors should be in R space. The sentence is corrected in the revised manuscript.
>
> > **Comment #13 (Dataset splits)** For the negative samples, how was the evaluation for Fig 2 done? How was the train/test split chosen since a random one likely doesn't model OOD effects?
> >
>
> In Fig. 2, the values in green are the ground truth from the GFP dataset (Sarkysian et al, 2016), the values in blue are predictions by an oracle trained without negative examples for random sequences, and the values in yellow are predictions by an oracle trained with negative examples for random sequences.
>
> We randomly split the data with a ratio of 90:10 for both GFP and His3. The initial states sampled by our optimization algorithm only use data from test sequences, so no data used during the training of the oracle and the training of the RL policy is used.
>
> Please note that, in the TAPE benchmark, the training data for GFP only includes sequences with 1-4 mutations, whereas the test data only includes sequences with 5 or more mutations. Such a split is helpful for evaluating the prediction of out of distribution sequences. We instead use a random split because our goal is to effectively increase our functionality predictor's trust region boundary in relation to the negative examples added during training. Section 4.1 has been revised to include information about the dataset splits.
>
> > **Comment #14** "CbAsoptimize"-> "CbAS optimize" on Page 5
> >
>
> Thanks for pointing it out. The typo is corrected in the revised manuscript.

---

> ### Author Response · Authors · 2022-11-17
> **Revised version of the manuscript uploaded**
>
> Dear Reviewer 62rv,
>
> We uploaded a revised highlighted version of the manuscript addressing the reviewer's comments. We would be happy to follow up on any feedback and to perform additional revision based on the reviewer's suggestions. Thank you very much for your attention and for the really detailed review that helped improve substantially the quality of our work.
>
> Sincerely, Authors6286

---

### Official Review · Reviewer_KJ1Y · 2022-10-24

**Confidence:** 4
**Correctness:** 2
**Technical Novelty And Significance:** 2
**Empirical Novelty And Significance:** 3
**Recommendation:** 3

**Clarity, Quality, Novelty And Reproducibility:**

The paper contains some nice detail in the models but it would be great to have an anonymous github, more detail about test/train splits, more detail on the datasets used would be useful for non domain experts. The pipeline needs to be better articulated too - I am not sure of soem of the notation
Related work: some key work seems to be missing eg Gomez-Bombarelli et al (2018) property prediction on the latent space for design, Notin 2022 Tranception - and results on GFP and HIs3,

**Strength And Weaknesses:**

Strengths:The approach combines existing models in a potentially interesting way : learning from the universe of natural proteins and labeled data from a large dataset of synthetic proteins with negative examples

Weaknesses:
 I believe the paper has a couple of  fundamental weaknesses as it stands that should be addressed before it could be accepted at ICLR. 1. While the evaluation is conceptually laudable  (performance, novelty, diversity), the metrics do not seem to capture the concepts well. Performance is evaluated irrespective of the distance from training.  Since ‘Novelty” is evaluated as binary- was in the training set or not. (I think - see equation in 3.1), the metric seems to have no measure of distance from training beyond a single mutation. (It is now well appreciated that predicting the effect of single mutations is relatively successful with even baseline methods such as conservation, Potts models or VAEs with alignments and transformers without alignments.) Therefore - for this piece of work to be evaluated I suggest it’s important to show sequence generation as a function of the distance from training data. A fundamental challenge in protein design is being able to generate sequences with a given function that have sequences different from natural or training examples. As one moves away from known sequences (in eg Hamming distance) - the harder it gets. For sequences that are only one  mutation away is relatively easy. ( many papers have shown this).  The performance results shown in Tables 1 indicate that their method is only 1% better than a  random single mutation for eg GFP, Table1, suggesting the metrics and/or the model is poor. Although the authors note this point , they do not follow up by addressing the reasons.
2. The use of he Oracle twice is circular - therefore invalidates the claims of performance; there are some ways around this that the authors could try
More minor weaknesses:
3. the reference used to justify the evaluation metrics is Hoffman et al 2022 - but this paper is about optimising small molecules - which are v different in "seq distance to function" relationships -  this is especially important in relation to the point about the Novelty measure above.
4. AlphaFold is not at all appropriate to support the claim of functional sequence optimisation - there are may mutations that will cause a protein to unfold that Alpha fold will predict as having almost exactly the same structure as it will align etc - therefore it proves nothing ( From their own FAQ page "AlphaFold has not been validated for predicting the effect of mutations. In particular, AlphaFold is not expected to produce an unfolded protein structure given a sequence containing a destabilising point mutation." And there are papers writing about this  eg Pak et al 2021


**Summary Of The Paper:**

This paper aims to generate protein sequences with high functionality and cellular fitness. The authors apply a reinforcement learning framework to walk through the space of latent representations of a pre-trained protein language model, using a functionality predictor as a reward function. This approach is tested for its ability to design examples of proteins GFP and His3 with predicted functionality, novelty, and diversity. The paper also tests improving the functionality predictor for use in reinforcement learning by including negative examples in training.


**Summary Of The Review:**

The paper has an interesting and somewhat novel approach but I am not convinced by the Performance and Novelty evaluation (as it is explained here) and I am uncertain about the AlphaFold result being informative as this would be the result of any single mutation even when known to destabilise the protein. This point about evaluation is non-trivial as papers like these should be setting the bar for future work; train test splits that are commensurate with the claims of the work and avoidance of circularity are really critical.

---

> ### Author Response · Authors · 2022-11-12
> **Response to reviewer KJ1Y [1/3]**
>
> The authors are incredibly thankful to the reviewers for their insightful and detailed comments that helped improve our work. Please find below our response to each comment. We would be happy to follow up on any additional feedback.
>
> > **Comment #1** **(Novelty metric)** Performance is evaluated irrespective of the distance from training. Since ‘Novelty” is evaluated as binary- was in the training set or not. (I think - see equation in 3.1), the metric seems to have no measure of distance from training beyond a single mutation. (It is now well appreciated that predicting the effect of single mutations is relatively successful with even baseline methods such as conservation, Potts models or VAEs with alignments and transformers without alignments.) Therefore - for this piece of work to be evaluated I suggest it’s important to show sequence generation as a function of the distance from training data. A fundamental challenge in protein design is being able to generate sequences with a given function that have sequences different from natural or training examples. As one moves away from known sequences (in eg Hamming distance) - the harder it gets. For sequences that are only one mutation away is relatively easy. ( many papers have shown this).
> >
>
> Thank you for your important suggestion. We now include a novelty metric based on a distance factor. The new metric “Novelty (distance)” is defined as the average distance between the generated sequence and sequences contained in the training dataset. In addition, the distance between the generated sequence and the wild-type is reported. This information will be added to the revised manuscript.
>
> In summary, our method achieves greater originality than the CbAS (Brookes et al., 2019). We note that (Swersky et al, 2020) can achieve higher distance-based novelty than our method and CbAS. However, this method cannot achieve high functionality and fitness values.
>
> |  | Performance | Novelty (Binary) | Novelty (Distance) | Distance from WT |
> | --- | --- | --- | --- | --- |
> | Ours | 3.531 ± 0.06 | 100% | 8.451±2.053 | 7.700±0.781 |
> | Baseline | 3.370 ± 0.013 | - | 7.704±2.658 | 6.849±1.905 |
> | CbAS (Brookes et al., 2019) | 3.328 ± 0.044 | 80% | 7.712±2.05 | 6.900±0.831 |
> | BO (Swersky et al., 2020) | 1.231 ± 0.034 | 100% | 36.961±5.511 | 36.700±5.311 |
>
> **Table** - Results obtained for GFP sequence design
>
> |  | Performance | Novelty (Binary) | Novelty (Distance) | Distance from WT |
> | --- | --- | --- | --- | --- |
> | Ours | 0.961 ± 0.050 | 60% | 8.361±2.077 | 10.950±1.322 |
> | CbAS (Brookes et al., 2019) | 0.889 ± 0.092 | 90% | 7.287±1.567 | 4.700±0.640 |
> | Baseline | 0.756 ± 0.013 | - | 6.889±1.572 | 6.710±1.567 |
> | BO (Swersky et al., 2020) | -0.089 ± 0.029 | 100% | 26.172±1.03 | 27.500±0.500 |
>
> **Table** - Results obtained for His3 sequence design
>
> > **Comment #2 (Performance metric)** The performance results shown in Tables 1 indicate that their method is only 1% better than a random single mutation for eg GFP, Table1, suggesting the metrics and/or the model is poor. Although the authors note this point , they do not follow up by addressing the reasons.
> >
>
> We appreciate your thoughtful comment. We apologize that we did not mention that log-fluorescence values were presented in Table 1. The difference between the proposed method (3.531) and the random single mutation baseline (3.410) can be interpreted as the average optimized sequence by the proposed method being 1.32 (=10^(3.531-3.410)) times brighter than the random single mutation baseline.
>
> > **Comment #3 (Oracle)** The use of the Oracle twice is circular - therefore invalidates the claims of performance; there are some ways around this that the authors could try.
> >
>
> Thank you for pointing out this critical issue. This point is very important since there might exist a leakage of information if we use the same oracle for optimization and evaluation. This concern was shared by other reviewers as well.
>
> One of the reviewers (62rv) suggested applying the method in (Kolli et al, 2022) and utilizing multiple oracles to ensure the reliability of results. This study suggests training multiple oracles with different numbers of layers, activation functions, and hyperparameters. We followed the advice and trained another oracle from scratch for the GFP dataset using an ESM model that uses different network parameters (35M). This oracle was not used for optimization.
>
> As a result, we could utilize two oracles: (i) an optimization oracle (trained using the ESM model with 150M parameters) which is used in the current work and (ii) an evaluation oracle (trained using the ESM model with 35M parameters). Please find the evaluation results based on this new oracle below:

---

> ### Author Response · Authors · 2022-11-12
> **Response to reviewer KJ1Y [2/3]**
>
> |  | Evaluation oracle | Optimization oracle |
> | --- | --- | --- |
> | Spearman’s Rho | 0.7426 | 0.8426 |
> | MSE | 0.2872 | 0.1436 |
> | Ours | 3.491 ± 0.352 | 3.531 ± 0.06 |
> | Baseline | 3.287 ± 0.237 | 3.370 ± 0.013 |
> | CbAS (Brookes et al., 2019) | 3.155 ± 0.153 | 3.328 ± 0.044 |
> | BO (Swersky et al, 2020) | 0.581 ± 0.095 | 1.231 ± 0.034 |
>
> **Table** - Results obtained for GFP sequence design
>
> |  | Evaluation oracle | Optimization oracle |
> | --- | --- | --- |
> | Spearman’s Rho | 0.6820 | 0.6635 |
> | MSE | 0.0110 | 0.0080 |
> | Ours | 0.945 ± 0.091 | 0.961 ± 0.050 |
> | Baseline | 0.616 ± 0.110 | 0.756 ± 0.013 |
> | CbAS (Brookes et al., 2019) | 0.749 ± 0.157 | 0.889 ± 0.092 |
> | BO (Swersky et al, 2020) | -0.313±0.065 | -0.089 ± 0.029 |
>
> **Table** - Results obtained for His3 sequence design
>
> Summarizing the results, we find that all methods show a decrease in performance using the new oracle. This decline was particularly pronounced for BO(Swesky et al, 2020). We also note that the standard deviation of the results increases with the new evaluation oracle.
>
> Our method continues to outperform all baselines under the new oracle. In addition to the results above, we examined the functionality values of top-K sequences generated by the proposed method. Two of the identified sequences had high functionality values of 3.9705265 and 3.8059266, indicating that they were 1.78 and 1.20 times brighter than the value for the wild type protein, respectively. In the future, we plan to evaluate these sequences through wet lab experiments.
>
> Thank you once again for this important feedback, and we will add the new results to the main paper for GFP and His3 datasets.
>
> > **Comment #4 (Reference for evaluation metrics)** The reference used to justify the evaluation metrics is Hoffman et al 2022 - but this paper is about optimising small molecules - which are v different in "seq distance to function" relationships - this is especially important in relation to the point about the Novelty measure above.
> >
>
> Thank you for this comment. We changed the main reference used to justify the evaluation metrics to Jain et al (2022). As suggested by the reviewer, we will also add novelty metrics based on distance and update the manuscript accordingly.
>
> > **Comment #5 (Validation using AlphaFold2)** AlphaFold is not at all appropriate to support the claim of functional sequence optimisation - there are may mutations that will cause a protein to unfold that Alpha fold will predict as having almost exactly the same structure as it will align etc - therefore it proves nothing ( From their own FAQ page "AlphaFold has not been validated for predicting the effect of mutations. In particular, AlphaFold is not expected to produce an unfolded protein structure given a sequence containing a destabilising point mutation." And there are papers writing about this eg Pak et al 2021
> >
>
> We agree with the reviewer that AlphaFold alone cannot be used to predict destabilizing point mutations. We nonetheless consider AlphaFold a valuable sanity check, as also mentioned by reviewer 62rv. AlphaFold can help verify sequences that are optimized far from the wild type protein, as is the case with our research.
>
> We believe the reviewer's comment opens up interesting future directions related to investigating how AlphaFold and the method explored in (Mansoor et al., 2022) can be used to provide additional support for functional sequence optimisation.
>
> [1] Mansoor, Sanaa, et al. "Accurate Mutation Effect Prediction using RoseTTAFold." *bioRxiv* (2022).
>
> > **Comment #6 (Reproducibility)** It would be great to have an anonymous github
> >
>
> We intend to make all the codes and implementation details available to the research community. We hope that by sharing the codes, our work can be reproduced and used by other researchers across various benchmarks.
>
> > **Comment #7 (Dataset split)** More detail about test/train splits
> >
>
> Thank you for this question, which we overlooked and it is very important. We randomly split the data with a ratio of 90:10 for both GFP and His3. The initial states sampled by our optimization algorithm only use data from test sequences, so no data used during the training of the oracle and the training of the RL policy is used.
>
> Please note that, in the TAPE benchmark, the training data for GFP only includes sequences with 1-4 mutations, whereas the test data only includes sequences with 5 or more mutations. Such a split is helpful for evaluating the prediction of out of distribution sequences. We instead use a random split because our goal is to effectively increase our functionality predictor's trust region boundary in relation to the negative examples added during training.
>
> The sentence in Section 4.1 is revised to include information about the dataset splits.

---

> ### Author Response · Authors · 2022-11-12
> **Response to reviewer KJ1Y [3/3]**
>
> > **Comment #8 (Methodology unclear)** The pipeline needs to be better articulated too - I am not sure of some of the notation
> >
>
> Thank you for the suggestion. We have revised the manuscript to the best of our ability. Please see the updated Methodology, Appendix A.1, and A.2 sections at the anonymized version linked at [https://bit.ly/3hBeS9E](https://bit.ly/3hBeS9E)
>
> > **Comment #9 (Related Work)** Some key work seems to be missing eg Gomez-Bombarelli et al (2018) property prediction on the latent space for design, Notin 2022 Tranception - and results on GFP and HIs3,
> >
>
> Thank you for suggesting these relevant research. We find them to be interesting and relevant to our research.
>
> In page 1, we include the previous work by Gomez-Bombarelli et al (2018) in the Introduction section as: “To tackle this problem in biological sequence design, previous literature (Gomez et al, 2018; Stanton et al, 2022) explored performing the optimization directly in a learned latent representation space. In this paper, we investigate the optimization of sequences via Reinforcement Learning in a latent representation space rather than in the protein sequence space”.
>
> In page 3, we include the previous work by Notin et al (2022) in the Related Works section as: “Inspired by transformers, Notion et al (2022) proposed a novel architecture named Tranception for autoregressive functionality prediction that achieves state-of-the-art performance in 87 benchmarks.”

---

> ### Author Response · Authors · 2022-11-17
> **Revised version of the manuscript uploaded**
>
> Dear Reviewer KJ1Y,
>
> We uploaded a revised highlighted version of the manuscript addressing the reviewer's comments. We would be happy to follow up on any feedback and to perform additional revision based on the reviewer's suggestions.
> Thank you very much for your attention and for the really detailed review that helped improve substantially the quality of our work.
>
> Sincerely,
> Authors6286

---

### Official Review · Reviewer_f1s7 · 2022-10-24

**Confidence:** 4
**Correctness:** 2
**Technical Novelty And Significance:** 3
**Empirical Novelty And Significance:** 2
**Recommendation:** 3

**Clarity, Quality, Novelty And Reproducibility:**

Clarity: There are major issues with the clarity of writing. It is difficult to judge other components as details about experimental setup, training procedures, etc. are missing. This needs to be addressed before publication.

**Strength And Weaknesses:**

Strength:
* The paper is theoretically sound. The issues identified with current approaches are real, and the proposed solutions are sensible.

Weaknesses:
* The paper is not carefully written and it was hard for me to find information about architecture, training setup, and experimental results. Examples:
    - The paper says the CLS token is used for an embedding of a sequence. Further it says average pooling is applied to this vector to reduce dimensionality. I am not familiar with average pooling being used for dimensionality reduction. The method should be explained or a citation provided.
    - While the paper says CLS tokens are used for dimensionality reduction, this is not clear from Figure 5.
    - The architecture of the MLP is not described in section A.1.
    - In section A.1, the paper states that encoder weights are not updated during fine-tuning, but the next sentence states that the encoder and decoder are jointly finetuned.
    - The decoder is not introduced in section A.1.
    - "Note that the information on mutated positions such as masks is not provided to the model." - I don't understand this statement. It would be nice to have a clearer explanation of the training procedure.
    - Details about training set split are not provided.
    - It was unclear how many total sequences the model was trained on and how many total sequences were seen in a given optimization round.
    - In section 4.3, it states that "only the latent vector [...] can simultaneously optimize GFP and His3 tasks." I'm not sure what this means -  how are the tasks being simultaneously optimized?
* The proposed method is initialized from sequences with relatively high functionality. The method proposed is able to optimize further from there. They also show a random mutation baseline that is similarly initialized. However, it does not seem that other methods are provided this initialization. This should be clearly discussed and some effort should be made to make this initialization.
* It's not clear what the highest functional value in the training set is. Is it higher than the values the model ultimately achieves?
* Is the encoder/decoder trained only on the training set or on the whole dataset?
* Figure 1 may benefit from providing a more detailed view of the process (what are the components trained on, how does sampling occur, etc.)

Minor Comments:
* The CLS token used for a protein-level vector but in ESM-2 it has no special meaning. Would averaging the final layers provide a better representation?
* Abstract: "GPF" -> GFP
* Page 5: "CbAsoptimize" -> "CbAs optimize"
* In tables, I personally dislike the use of "Baseline" as an entry. Especially given that there are other methods you compare against, it would be nice to clearly define what this is in the table.


**Summary Of The Paper:**

The paper proposes a model-based reinforcement learning method for sequence design. The main innovation is that design is performed in the latent space, which should allow easier optimization given the combinatorial search space. Furthermore, the authors point out the flaw of using oracles trained on experimental data as the sole source of truth, as they give high functional scores to random sequences. They demonstrate that adding random sequences to the training set alleviates this issue.

**Summary Of The Review:**

The paper proposes a theoretically sound procedure, but many small details are missing that make it difficult to assess the results. Clearer explanations of all aspects of training, model architecture, and experiments are needed. Additionally, it is possible the experimental setting is unfairly advantaging the proposed method by initializing from high-function sequences. Some attempts should be made to provide similar initialization to other methods, or it should be explained why this is not possible.

---

> ### Author Response · Authors · 2022-11-12
> **Response to reviewer f1s7 [1/3]**
>
> The authors are incredibly thankful to the reviewers for their insightful and detailed comments that helped improve our work. Please find below our response to each comment. We would be happy to follow up on any additional feedback.
>
> ### Unclear languages in methodology
>
> The authors appreciate the reviewer's critique of the unclear language in the Methodology and Appendix A.1 and A.2 sections. In the revised manuscript, we tried to conduct a comprehensive review. We rewrote the corresponding sections to address the reviewer's concerns and enhance the clarity of the experimental design. Please see the updated version of the Methodology, Appendix A.1, Appendix A.2, and Fig. 5 at [https://bit.ly/3hBeS9E](https://bit.ly/3hBeS9E)
>
> > **Comment #1** The paper says the CLS token is used for an embedding of a sequence. Further it says average pooling is applied to this vector to reduce dimensionality. I am not familiar with average pooling being used for dimensionality reduction. The method should be explained or a citation provided.
> >
>
> We are happy to answer this question. Average pooling is commonly used in computer vision tasks to reduce dimensionality. Because protein engineering requires generating an embedding with a fixed size for sequences of varying lengths, average pooling or similar strategies are viable options.
>
> In natural language processing, average pooling has been used to provide a 'summary' or 'context' of the embeddings for a similar problem involving summarizing the embedding for variable-length sentences [1,2].
>
> In our case, we use 1-Dimensional adaptive average pooling implemented in Pytorch [3] to reduce from (E,) to (R,). We apologize that our description of this process was unclear. We hope the newly revised version clearly states these points in the Methodology.
>
> [1] Zhao, Shuai, et al. "AP-BERT: enhanced pre-trained model through average pooling." Applied Intelligence (2022): 1-9.
>
> [2] Chen, Qian, Zhen-Hua Ling, and Xiaodan Zhu. "Enhancing sentence embedding with generalized pooling." arXiv preprint arXiv:1806.09828 (2018).
>
> [3] https://pytorch.org/docs/stable/generated/torch.nn.AdaptiveAvgPool1d.html
>
> > **Comment #2** While the paper says CLS tokens are used for dimensionality reduction, this is not clear from Figure 5.
> >
>
> Fig. 5 is redrawn to accommodate the reviewer's suggestion and to provide additional information to clarify the architecture of the proposed method.
>
> > **Comment #3** The architecture of the MLP is not described in section A.1.
> >
>
> Thank you for this feedback. Now we add more details about the MLP architecture on Fig. 5 and Methodology.
>
> > **Comment #4** In section A.1, the paper states that encoder weights are not updated during fine-tuning, but the next sentence states that the encoder and decoder are jointly finetuned.
> >
>
> Thank you for pointing this out. The reviewer is correct that only the decoder weights are modified during fine-tuning. The encoder weights are fixed. The correction is made in the revised manuscript.
>
> > **Comment #5** The decoder is not introduced in section A.1.
> >
>
> The decoder architecture was described in the Methodology section. Now, we provide additional information about the decoder in section A.1, including additional details in Fig. 5.
>
> > **Comment #6** "Note that the information on mutated positions such as masks is not provided to the model." - I don't understand this statement. It would be nice to have a clearer explanation of the training procedure.
> >
>
> Based on the suggestion, we are now including additional information regarding the encoder and decoder training procedure. We hope the message is clear that even though the information on the mutated positions is used to calculate the loss function, this information is not given to the model as input.
>
> > **Comment #7** Details about training set split are not provided.
> >
>
> Thank you for this question, which we overlooked and it is very important.
>
> We randomly split the data with a ratio of 90:10 for both GFP and His3. The initial states sampled by our optimization algorithm only use data from test sequences, so no data used during the training of the oracle and the training of the RL policy is used. Please note that, in the TAPE benchmark, the training data for GFP only includes sequences with 1-4 mutations, whereas the test data only includes sequences with 5 or more mutations. Such a split is helpful for evaluating the prediction of out of distribution sequences.
>
> We instead use a random split because our goal is to effectively increase our functionality predictor's trust region boundary in relation to the negative examples added during training. The sentence in Section 4.1 is revised to include information about the dataset splits.

---

> > ### Comment · Reviewer_f1s7 · 2022-11-21
> > **Response to authors**
> >
> > Thank you for updating the paper in response to my comments.
> >
> > While I think the paper has improved, I still have significant issues with components of the paper.
> >
> > 1. My primary concern is the training of the off-policy algorithm. The evaluations and results all show the performance of the _trained_ off-policy RL algorithm, but how is this algorithm trained in the first place? How many samples does it use and where does that training data come from? What is the highest functional value in the training data of the off-policy algorithm? Is it higher or lower than the results the model ultimately achieves?
> > 2. I am broadly concerned with a setup in which a random split is used to train _any_ portion of the model other than the evaluation oracle. This is because the distribution of sequences in the dataset may not be random (in the case of GFP, for example, more sequences have 2-3 mutations than higher numbers of mutations). This may leak information about which sequences are likely to be functional (there are ways of incorporating similar information, e.g. by using an MSA, that would be more reasonable). In the case of the proposed method, the problem is worse as the functionality predictor is trained using functional measurements, then these provide reward to the off-policy optimizer. Therefore, it seems to me that the off-policy optimizer is provided hidden information about the distribution.
> >
> > Minor Comment: I am aware of the use of average pooling to reduce _spatial_ or _temporal_ dimensions (as in computer vision) but here you are using it to reduce the embedding dimension. This is not necessarily invalid, but it is a highly unusual use of average pooling (training a linear projection or using PCA is much more common for this use case). Note that the embedding dimension is fixed, so there should be no need to use adaptive reduction methods.

---

> > > ### Comment · Reviewer_f1s7 · 2022-11-21
> > > **Further comment**
> > >
> > > Stepping back a little, I think there is a mismatch between the goals of the paper and the execution.
> > >
> > > *Goal (as I understand):* Optimizing in the latent space allows you to take steps along the manifold of functional protein sequences (as understood by the sequence encoder). This allows you to take larger steps in sequence space and be confident that steps are likely to take you to functional sequences.
> > >
> > > However, this is implemented by training a sequence encoder/decoder on the sequences from the experiment. This has both positive *and* negative biases. Now, steps do not take you on the manifold of functional protein sequences - they take you on the manifold of "sequences that were in this particular experiment". This includes high functionality sequences (potentially leading to overperformance) and non-functional sequences (potentially leading to underperformance). For example, if you trained on a dataset of saturation mutagenesis data, containing every possible mutation at all positions, I think you would expect steps in the latent space to be no better than random sampling!
> > >
> > > The right thing to do then is to train on data which you believe will keep you on the manifold of functional sequences. This may be data from arbitrary protein sequences (e.g. making mutations on sequences in Uniprot) or on data in the evolutionary neighborhood of a sequence of interest (i.e. the MSA data).

---

> ### Author Response · Authors · 2022-11-12
> **Response to reviewer f1s7 [2/3]**
>
> > **Comment #8** It was unclear how many total sequences the model was trained on and how many total sequences were seen in a given optimization round.
> >
>
> We have divided our datasets into train and test sets. The train set is used to train the functionality predictor and encoder/decoder. The test set is only used to sample initial states in the optimization round. For evaluation, 100 sequences from the test set, following the methodology described in Appendix A.5, are sampled and optimized by the proposed method.
>
> > **Comment #9** In section 4.3, it states that "only the latent vector [...] can simultaneously optimize GFP and His3 tasks." I'm not sure what this means - how are the tasks being simultaneously optimized?
> >
>
> We appreciate your feedback. To eliminate ambiguity, the word "simultaneously" is removed from the sentence in the revised manuscript.
>
> ### Other comments
>
> > **Comment #10 (Initialization)** The proposed method is initialized from sequences with relatively high functionality. The method proposed is able to optimize further from there. They also show a random mutation baseline that is similarly initialized. However, it does not seem that other methods are provided this initialization. This should be clearly discussed and some effort should be made to make this initialization.
> >
>
> We appreciate your feedback. BO (Swesky et al, 2022) and CbAS (Brookes et al, 2019) are initialized from high functional sequences that are within the same range as the initial states for the proposed method, as shown in Appendix A.5. DynaPPO (Angermueller et al, 2019) generates the entire sequence amino acid by amino acid, so there is no initialization. The other generative method, GFlowNet (Jain et al, 2022), generates sequences so there is no initialization. We provide a full train split of the dataset as the initial dataset to train their proxy block.
>
> As the reviewer points out, our method, random mutations, BO, and CbAS all benefit from the initialization method. As this is a crucial issue for validating the results and comparisons presented in our manuscript, we will provide additional information on the initialization of the optimization process for the comparison baseline methods.
>
> > **Comment #11** It's not clear what the highest functional value in the training set is. Is it higher than the values the model ultimately achieves?
> >
>
> The highest functionality value in the training set for the GFP dataset is 4.12, while the functionality value for the wild type is 3.72. For the His3 dataset, the functionality is relative growth rate to the wild type so the wild-type fitness value is equal to 1 and the maximum fitness value is equal to 1.63. Our average performance is higher than the initial state but still is lower than the value of the wild type.
>
> While preparing our response to Comment #3 by Reviewer #3 (KJ1J), we observed that two of the optimized sequences led to predicted functionality values higher than the wild-type by using a new evaluation oracle not used for optimization.
>
> > **Comment #12** Is the encoder/decoder trained only on the training set or on the whole dataset?
> >
>
> The encoder, the decoder, and the functionality predictor are trained solely on the training set. The only purpose of the test set is to sample initial states.
>
> > **Comment #13** Figure 1 may benefit from providing a more detailed view of the process (what are the components trained on, how does sampling occur, etc.)
> >
>
> We augmented Fig. 1 with dimensions and revised the caption. We also updated Fig. 5, Methodology, and Appendix to the best of our ability for clarity. Please see the updated version of the Methodology, Appendix A.1, Appendix A.2, and Fig. 5 at [https://bit.ly/3hBeS9E](https://bit.ly/3hBeS9E)
>
> The revised caption of Fig. 1 is: “Overview of the proposed framework. Top Left depicts the encoder-decoder architecture trained to represent protein sequences in the latent space. Top Right depicts the RL framework. The state is defined as the representation in the latent space computed by the encoder and the action is defined as a perturbation in this representation. The perturbed representation is decoded back to a protein sequence that is evaluated using a functionality predictor. The predicted functionality is used to compute the final reward. Bottom depicts three possible ways to model state and action in RL-based biological sequence design.”

---

> ### Author Response · Authors · 2022-11-12
> **Response to reviewer f1s7 [3/3]**
>
> ### Minor comments
>
> > **Minor Comment #1** **(CLS Token)** The CLS token used for a protein-level vector but in ESM-2 it has no special meaning. Would averaging the final layers provide a better representation?
> >
>
> Thank you for the insightful comment. Our initial goal was to generate the per-sequence embeddings via averaging, as shown in the template given by the official repository of ESM [1].
>
> However, during our preliminary exploration experiments with this architecture, we discovered that utilizing only the embeddings from the CLS token (i.e., the first token in [1]) produced superior results when compared to the averaging method. We conjecture this may be because CLS tokens contain special information, such as the sequence's aggregate representation.
>
> [1] https://github.com/facebookresearch/esm
>
> > **Minor Comment #2 (Typo)** Abstract: "GPF" -> GFP
> >
>
> The typo is corrected in the revised manuscript. Thank you!
>
> > **Minor Comment #3 (Typo)** Page 5: "CbAsoptimize" -> "CbAs optimize"
> >
>
> The typo is corrected in the revised manuscript. Thank you!
>
> > **Minor Comment #4 (Naming)** In tables, I personally dislike the use of "Baseline" as an entry. Especially given that there are other methods you compare against, it would be nice to clearly define what this is in the table.
> >
>
> We agree that the term "Baseline" can be misleading. In the revised manuscript we substitute the term with “Directed Evolution”. In addition, we provide an explanation of how this result was obtained.

---

> ### Author Response · Authors · 2022-11-17
> **Revised version of the manuscript uploaded**
>
> Dear Reviewer f1s7,
>
> We uploaded a revised highlighted version of the manuscript addressing the reviewer's comments. We would be happy to follow up on any feedback and to perform additional revision based on the reviewer's suggestions.
> Thank you very much for your attention and for the really detailed review that helped improve substantially the quality of our work.
>
> Sincerely,
> Authors6286

---

### Official Review · Reviewer_F4US · 2022-10-25

**Confidence:** 4
**Correctness:** 4
**Technical Novelty And Significance:** 3
**Empirical Novelty And Significance:** 3
**Recommendation:** 8

**Clarity, Quality, Novelty And Reproducibility:**

## Clarity
The paper is clearly written, with excellent figures to support the story.

## Quality
The quality of the paper is high.

## Reproducibility
The authors have not made source code available as part of their submission, making it difficult to fully assess reproducibility. The method is, however, presented at a level of detail that should make it possible to reproduce the results. I strongly encourage the authors to share their source code upon acceptance of their paper.

## Detailed comments

Page 1. "To tackle this problem, in this paper we propose..."
This paragraph, and especially this sentence, suggests that latent space optimization of proteins is new, ignoring recent work on Bayesian Optimization of proteins on latent spaces, such as "Accelerating Bayesian Optimization for Biological Sequence Design with Denoising Autoencoders" (ICML 2022). This paragraph should therefore be rephrased.
This same ICML2022 reference should also be added to the "Biological sequence design" subsection on the next page, and ideally compared to in the results section. I am not affiliated with this paper in any way - but merely suggesting it because the methods are closely related, and both are contestants to constitute the current state of the art - so comparing them head-on would be relevant to the community.

Page 3: "k is predicted from a representation q with dimensions (L, E), and". Page 4: "As a result of performing the action a_t, the agent receives the reward r_t.". Page 5: "a dense reward, defined as r_t= f(s_t)".
The last two statements seem to be at odds with the first statement. Originally, f is described as acting on a q representation of size (L,E), but later, the reward is calculated based on s_t, which is only E-dimensional. Please clarify.

Page 4. "The dataset proposed in (Sarkisyan et al., 2016) is used to train the GFP encoder-decoder and its functionality predictor."
It would be helpful if you could discuss somewhere in the paper how many parameters are involved in estimating the task-specific projection to lower dimensions (I assume that this is just ExR) and whether this estimation becomes problematic for smaller datasets that the one you studied here. Likewise for the oracle - although one could presumably use a general oracle instead of a task specific one in this case.

Page 4. "Datasets were split into train and test sets." How? Just uniformly?

Page 5. "We compare with four optimization methods"
I assume these are all optimized on the same oracle(?). As mentioned earlier, it would be beneficial to the community if you compared directly to the ICML2022 method here as well - if at all possible.

Page 6. "The sequence alone cannot easily convey structural information about a protein’s functional sites, making it unsuitable for guiding the optimization process."
I'm surprised by these results. In Table 2, you show that Random mutations work well as an optimization strategy on His3. Why does it fail completely in the experiment in Table 3? Wouldn’t you at least expect to recover the most important sites in the protein? Does this result perhaps primarily indicate a weakness in your policy optimization rather than the representation itself?

Page 7. "...is compared to random perturbation and BO,"
When writing “BO”, are you referring to a general Bayesian Optimization procedure - or the specific one that you cite earlier (Swersky et al., 2020)? Since you are using a continuous latent space here, I assume this is now a different BO - in which case you should make the distinction clear - and explain what the setup is - is it a standard GP-based BO with a expected improvement acquisition function? I would also suggest that the authors make it clear that this is just one particular (and perhaps particularly simple) choice of BO.

Page 7. "Fig. 2 shows that the functionality predictor trained without negative examples incorrectly predicts a high value for non-functional sequences (mean=4.002)"
Isn’t it odd that the oracle predicts a higher average functional value than any value it has observed during training? Does this indicate something is flawed with the oracle?

Page 7. "Fig. 3(d) presents optimization steps taken by the trained policy, which now shows that large (optimistic) perturbations taken by the policy often lead to failure in optimization."
It was not quite clear to me what this figure is meant to illustrate. Are the large steps failure modes of the learned policy? (i.e. should it have learned to prevent such steps?)

Page 7. Discussion
It would be helpful if the authors somewhere in the paper discussed the source of improvement over e.g. Bayesian Optimization. The BO and reinforcement learning literatures have quite different terminology and it can be a bit difficult to see exactly which components make a difference in practice.
Is it the fact that a policy is learned vs the fixed acquisition function typically used in a BO setting?


### Minor

Page 1. "The first cause is that the optimization process is usually performed by generating candidate sequences through amino acid substitutions"
Slightly odd statement, since any method (including the one proposed here), will ultimately use amino acid substitutions (or insertions/deletion). Perhaps writing "sequences *directly* through amino acid substitutions" would be better?

Page 3. "a 2-dimensional vector representation q ∈ Q of dimensions (L, E)"
Slightly confusion that the representation is both 2 dimensional and has dimension L,E. Consider rephrasing.

Page 4. "Negative examples are defined as random sequences with a zero functionality value."
Do you standardize the functionality values in any way? Otherwise, 0 seems like an arbitrary value.

Page 5. "We set three experimental rewards". At this point in the text it is not clear whether these losses will be used simultaneously or as alternatives. Perhaps add "alternative" here to make this clear.

Page 5. "The performance evaluation metric is calculated as the mean log-fluorescence intensity from the top K generated sequences."
For clarify, perhaps make it clear that this is according to the oracle, and not the ground truth values.

Page 5. "CbAsoptimize"
Missing space

**Strength And Weaknesses:**

The paper is well written, the method well described, and the results are convincing. Please see below for detailed comments to specific parts of the paper.

**Summary Of The Paper:**

The manuscript presents a procedure for protein engineering using a model-based reinforcement approach building on the ESM2 language model. The authors demonstrate that the high dimensional ESM2 representation can be mapped to a lower dimensional representation space which is suitable for optimization, and that full amino acid sequences can be reconstructed from this reduced representation with meaningful accuracy. They then propose a off-policy reinforcement learning procedure to learn how to make updates in the representation space. Finally, the authors evaluate the method on two protein engineering datasets, and report convincing results.

**Summary Of The Review:**

The paper proposes a new method for protein engineering. It is well-written, carefully documents its claims, and demonstrates convincing results. I have only minor suggestions for edits to the paper.

---

> ### Author Response · Authors · 2022-11-12
> **Response to reviewer F4US [1/4]**
>
> The authors are incredibly thankful to the reviewers for their insightful and detailed comments that helped improve our work. Please find below our response to each comment. We would be happy to follow up on any additional feedback.
>
> > **Comment #1** The authors have not made source code available as part of their submission, making it difficult to fully assess reproducibility.
> >
>
> We intend to make all the codes and implementation details available to the research community. We hope that by sharing the codes, our work can be reproduced and used by other researchers across various benchmarks.
>
> > **Comment #2** Page 1. "To tackle this problem, in this paper we propose..." This paragraph, and especially this sentence, suggests that latent space optimization of proteins is new, ignoring recent work on Bayesian Optimization of proteins on latent spaces, such as "Accelerating Bayesian Optimization for Biological Sequence Design with Denoising Autoencoders" (ICML 2022). This paragraph should therefore be rephrased. This same ICML2022 reference should also be added to the "Biological sequence design" subsection on the next page, and ideally compared to in the results section. I am not affiliated with this paper in any way - but merely suggesting it because the methods are closely related, and both are contestants to constitute the current state of the art - so comparing them head-on would be relevant to the community.
> >
>
> Thank you very much for introducing us to this research. We find the work to be relevant to our research. Below, we summarize the key differences between our work and the introduced research.
>
> First, the optimization paradigm in Stanton et al (2022) employs the concept of corrupt-and-denoise the sequence, whereas we optimize from the embeddings of the sequence that is considered an initial state. Our process allows the intermediate states in the latent space to be decoded back into sequences.
>
> Second, we define optimization as an episodic task and employ reinforcement learning. In contrast, Stanton et al (2022) uses Bayesian optimization and a multi-objective acquisition function.
>
> Based on the suggestion, we discuss this research in Introduction (page 1) and in Related Work (page 2). For example, we added the following:
>
> In page 1, we rephrase the sentence about our contribution as: “To tackle this problem in biological sequence design, previous literature (Gomez et al, 2018; Stanton et al, 2022) explored performing the optimization directly in a learned latent representation space. In this paper, we investigate the optimization of sequences via Reinforcement Learning in a latent representation space rather than in the protein sequence space”.
>
> In page 2 (on “Biological sequence design”), we added the explanation and comparison with Stanton et al (2022): “Recently, Stanton et al. (2022) proposed the use of Bayesian optimization on a latent space for drug design. The method trains a denoising autoencoder that is used to generate embeddings for corrupted sequences. The embeddings are optimized using a multi-objective acquisition function and the sequence then recovered. It is important to note that our work does not use a corrupt-and-denoise concept, and define optimization as an episodic task to use reinforcement learning algorithms.”
>
> During the discussion period, we will also try our best to report experimental comparison with Stanton et al (2022).
>
> > **Comment #3 (Clarification of formulation)** Page 3: "k is predicted from a representation q with dimensions (L, E), and". Page 4: "As a result of performing the action a_t, the agent receives the reward r_t.". Page 5: "a dense reward, defined as r_t= f(s_t)". The last two statements seem to be at odds with the first statement. Originally, f is described as acting on a q representation of size (L,E), but later, the reward is calculated based on s_t, which is only E-dimensional. Please clarify.
> >
>
> Thank you for pointing out this mistake. We updated Fig. 5 and changed the writing in the Methodology part to clarify these statements; the anonymized version can be found at https://bit.ly/3hBeS9E. We rewrote these statements to consider the input of the functionality predictor as being the sequence x. When we calculate the reward, we first decode the sequence from s_t, and then use the decoded sequence as input for the functionality predictor.

---

> ### Author Response · Authors · 2022-11-12
> **Response to reviewer F4US [2/4]**
>
> > **Comment #4 (Number of Parameters and Smaller Datasets)** Page 4. "The dataset proposed in (Sarkisyan et al., 2016) is used to train the GFP encoder-decoder and its functionality predictor." It would be helpful if you could discuss somewhere in the paper how many parameters are involved in estimating the task-specific projection to lower dimensions (I assume that this is just ExR) and whether this estimation becomes problematic for smaller datasets that the one you studied here. Likewise for the oracle - although one could presumably use a general oracle instead of a task specific one in this case.
> >
>
> Thank you for your insightful comments. We rewrote Appendix A.1 and updated Figure 5 to the best of our abilities to increase clarity and provide more details; the anonymized version can be found at [https://bit.ly/3hBeS9E](https://bit.ly/3hBeS9E)
>
> Regarding the feedback on small datasets, even though we fine-tune a language model trained on sequences from the protein universe, this estimation may be problematic. This presents an intriguing challenge; using a general oracle and an unified optimization policy is an interesting approach to mitigate this issue. We appreciate your input and hope to investigate this further in the future.
>
> > **Comment #5 (Dataset split)** Page 4. "Datasets were split into train and test sets." How? Just uniformly?
> >
>
> Thank you for this question, which we overlooked and it is very important. We randomly split the data with a ratio of 90:10 for both GFP and His3. The initial states sampled by our optimization algorithm only use data from test sequences, so no data used during the training of the oracle and the training of the RL policy is used.
>
> Please note that, in the TAPE benchmark, the training data for GFP only includes sequences with 1-4 mutations, whereas the test data only includes sequences with 5 or more mutations. Such a split is helpful for evaluating the prediction of out of distribution sequences. We instead use a random split because our goal is to effectively increase our functionality predictor's trust region boundary in relation to the negative examples added during training.
>
> Section 4.1 has been revised to include information about the dataset splits.
>
> > **Comment #6 (Oracle)** Page 5. "We compare with four optimization methods" I assume these are all optimized on the same oracle(?)
> >
>
> Yes, the reviewer is correct in that, in our experiment, all methods were optimized using the same oracle trained on negative samples. This design choice was made specifically to investigate the use of negative samples in increasing the oracle's robustness.
>
> > **Comment #7 (Optimization in Sequence Space)** Page 6. "The sequence alone cannot easily convey structural information about a protein’s functional sites, making it unsuitable for guiding the optimization process." I'm surprised by these results. In Table 2, you show that Random mutations work well as an optimization strategy on His3. Why does it fail completely in the experiment in Table 3? Wouldn’t you at least expect to recover the most important sites in the protein? Does this result perhaps primarily indicate a weakness in your policy optimization rather than the representation itself?
> >
>
> Thank you for spotting a mistake in our action definition for the ablation study in Table 3. The purpose of Table 3 was to evaluate the influence of the state and action modeling on training the RL policy. For that, we compared with an autoregressive generation of the protein sequence using the conditional formulation of (Angermueller et al., 2019) and a multi-discrete formulation generating the entire sequence. Thus, the results could demonstrate the difficulty in generating the entire discrete amino acid sequence during optimization.
>
> We corrected Section 4.3 from “We investigate three types of action modeling: (i) a mutation in the amino acid sequence, […]” to “We investigate three types of action modeling: (i) multi discrete sequence generation, (ii) conditional autoregressive addition of amino acids proposed in (Angermueller et al, 2019) […]”.
>
> We also rewrote the statement about the results of using the one-hot encoding of the sequence as input as follows: “Compared to the latent space learned based on the embeddings from ESM-2, using the one-hot encoded mutant sequence as input makes the identification of structural changes related to the protein’s functional site challenging.”
>
> Thank you once again for the comment, and we plan to perform an analysis in which the RL policy chooses the position and amino acid to mutate as its action modeling in the future.

---

> ### Author Response · Authors · 2022-11-12
> **Response to reviewer F4US [3/4]**
>
> > **Comment #8 (Clarification regarding Bayesian Optimization Methods)** Page 7. "...is compared to random perturbation and BO," When writing “BO”, are you referring to a general Bayesian Optimization procedure - or the specific one that you cite earlier (Swersky et al., 2020)? Since you are using a continuous latent space here, I assume this is now a different BO - in which case you should make the distinction clear - and explain what the setup is - is it a standard GP-based BO with a expected improvement acquisition function? I would also suggest that the authors make it clear that this is just one particular (and perhaps particularly simple) choice of BO.
> >
>
> The reviewer is correct to point out that BO on Page 7 refers to the method proposed by (Swersky et al, 2020). We have revised the corresponding text in Page 7 to make this point clear. We now provide more details about the setup and optimization process for BO used in the ablation study.
>
> > **Comment #9 (Problem with Oracle trained without negative examples)** Page 7. "Fig. 2 shows that the functionality predictor trained without negative examples incorrectly predicts a high value for non-functional sequences (mean=4.002)" Isn’t it odd that the oracle predicts a higher average functional value than any value it has observed during training? Does this indicate something is flawed with the oracle?
> >
>
> We appreciate this question. This figure was used to demonstrate how an oracle trained without negative examples may produce incorrect results. We hence propose that an oracle be carefully trained and verified for reliable model-based optimization; this is the central message of our work. Using negative examples to create "a trust region" for the oracle's predictions is one such strategy, as proposed in this work. We envision there could be other strategies, such as the use of different normalization and non-linear activation functions. We will revise the paper to improve the clarity of this message.
>
> > **Comment #10 (Failure optimization cases with His3)** Page 7. "Fig. 3(d) presents optimization steps taken by the trained policy, which now shows that large (optimistic) perturbations taken by the policy often lead to failure in optimization." It was not quite clear to me what this figure is meant to illustrate. Are the large steps failure modes of the learned policy? (i.e. should it have learned to prevent such steps?)
> >
>
> Throughout the evaluation, we generated many sequences in which the top-K (highest functionality/fitness values) are compared.
>
> In Fig. 3(d), we also investigated sequences that were generated by our policy and were associated with low functionality/fitness values (failure modes of the learned policy). Desirably, the policy should have learned to prevent such cases. We observed that, for the His3 dataset-trained optimization policy, large perturbations sometimes resulted in the policy being at the bottom of the valley. These cases present difficulties for optimization techniques.
>
> We are currently investigating exploration approaches to achieve successful optimization even from sequences associated with low functionality/fitness values, which should be critical for the design of de novo proteins with high functionality/fitness values.
>
> > **Comment #11 (Reinforcement Learning vs Bayesian Optimization)** Page 7. Discussion It would be helpful if the authors somewhere in the paper discussed the source of improvement over e.g. Bayesian Optimization. The BO and reinforcement learning literatures have quite different terminology and it can be a bit difficult to see exactly which components make a difference in practice. Is it the fact that a policy is learned vs the fixed acquisition function typically used in a BO setting?
> >
>
> Thank you for this interesting suggestion. Comparing Bayesian Optimization (BO) and Reinforcement Learning (RL) in depth and examining theoretical differences in optimization could be valuable.
>
> We will try our best to follow up on this suggestion during the paper discussion phase. We plan to include additional discussion regarding the sources of improvement of RL over BO related to differences in the modeling of the problem setting (e.g. episodic task), the optimization algorithm, and the exploration strategies.

---

> ### Author Response · Authors · 2022-11-12
> **Response to reviewer F4US [4/4]**
>
> ### Minor comments
>
> > **Minor** **Comment #1** Page 1. "The first cause is that the optimization process is usually performed by generating candidate sequences through amino acid substitutions" Slightly odd statement, since any method (including the one proposed here), will ultimately use amino acid substitutions (or insertions/deletion). Perhaps writing "sequences *directly* through amino acid substitutions" would be better?
> >
>
> According to the reviewer’s suggestion, the corresponding sentence is rewritten using the word “directly” in the revised manuscript.
>
> > **Minor** **Comment #2** Page 3. "a 2-dimensional vector representation q ∈ Q of dimensions (L, E)" Slightly confusion that the representation is both 2 dimensional and has dimension L,E. Consider rephrasing.
> >
>
> According to the reviewer’s suggestion, we rephrase the sentence to solve the incorrect statement.
>
> > **Minor** **Comment #3** Page 4. "Negative examples are defined as random sequences with a zero functionality value." Do you standardize the functionality values in any way? Otherwise, 0 seems like an arbitrary value.
> >
>
> The GFP value of 0 denotes an extremely low fluorescence level (10^(0-3.72)=0.0002 times the brightness of the wild-type protein). The His3 value of 0 denotes the minimum fitness value, ranging from [0,1.63]. The reviewer is correct in that the value chosen when training the oracle will be determined by the dataset and the standardization of the functionality/fitness values.
>
> We rephrased the sentence to "Negative examples are defined as random sequences with a low functionality/fitness value." In addition, the revised manuscript now includes the clarification contained in this response regarding the definition of negative samples in Appendix A.2.
>
> > **Minor** **Comment #4** Page 5. "We set three experimental rewards". At this point in the text it is not clear whether these losses will be used simultaneously or as alternatives. Perhaps add "alternative" here to make this clear.
> >
>
> Thank you for this feedback. We have added the word “alternative” to clarify the reward definition and its usage.
>
> > **Minor** **Comment #5** Page 5. "The performance evaluation metric is calculated as the mean log-fluorescence intensity from the top K generated sequences." For clarify, perhaps make it clear that this is according to the oracle, and not the ground truth values.
> >
>
> We appreciate this comment. Section 4.1 (Evaluation Metrics) has been modified to clearly describe that the performance evaluation metrics is calculated using an oracle because we do not have ground truth values for sequences generated that are not in the training/test sets. Below is the revised text:
>
> “The performance evaluation metric is calculated as the mean predicted log-fluorescence intensity from the top K generated sequences. The predicted log-fluorescence intensity is obtained by using a functionality predictor trained using the methodology presented in Section 3.2.”
>
> > **Minor** **Comment #6** Page 5. "CbAsoptimize" Missing space
> >
>
> Thanks for pointing it out. The typo is corrected in the revised manuscript.

---

> ### Author Response · Authors · 2022-11-17
> **Revised version of the manuscript uploaded**
>
> Dear Reviewer F4US,
>
> We uploaded a revised highlighted version of the manuscript addressing the reviewer's comments. We would be happy to follow up on any feedback and to perform additional revision based on the reviewer's suggestions.
> Thank you very much for your attention and for the really detailed review that helped improve substantially the quality of our work.
>
> Sincerely,
> Authors6286

---

### Decision · Program_Chairs · 2023-01-20

**Decision:**

Reject

**Justification For Why Not Higher Score:**

Overall the reviewers are reasonably consistent in terms of their assessment on the strengths and weaknesses of this work. It is very commendable that the authors have improved their work during the discussion, but major fundamental concerns, particularly as noted by reviewers f1s7, KJ1Y, 62rv, remain. The authors also mentioned that they are expanding the datasets for benchmarking and it would be very helpful for authors to take these steps to enhance the results and robustness of their evaluation/comparison. These improvements will make this work a strong candidate for a future submission.

**Justification For Why Not Lower Score:**

N/A

**Metareview: Summary, Strengths And Weaknesses:**

This work proposed an approach for protein sequence design and engineering, using a framework extending from existing language model, which generate a low-dimensional latent space that the authors could leverage for sequence optimization. The use an off-policy RL method to help update designs in the representation space. And the authors demonstrate the method's performance using two protein variant datasets and reported good performance compared to baseline and existing methods. Overall the ideas proposed in this work are very good directions to help inspire better methods in the area of protein engineering and sequence design, as noted by several reviewers. In addition, the presentation of the work and results are overall clear and help to showcase the importance of main ideas. The main weaknesses are that: without a broader and deeper benchmarking and metric justification, it would be not straightforward to evaluate the improvement of the proposed method in a convincing fashion.